# Robust Compressed Sensing using Generative Models

**Ajil Jalal** *
ECE, UT Austin
ajiljalal@utexas.edu

**Liu Liu**
ECE, UT Austin
liuliu@utexas.edu

**Alexandros G. Dimakis**
ECE, UT Austin
dimakis@austin.utexas.edu

**Constantine Caramanis**
ECE, UT Austin
constantine@utexas.edu

## Abstract

The goal of compressed sensing is to estimate a high dimensional vector from an underdetermined system of noisy linear equations. In analogy to classical compressed sensing, here we assume a generative model as a prior, that is, we assume the vector is represented by a deep generative model $G : \mathbb{R}^k \to \mathbb{R}^n$. Classical recovery approaches such as empirical risk minimization (ERM) are guaranteed to succeed when the measurement matrix is sub-Gaussian. However, when the measurement matrix and measurements are heavy-tailed or have outliers, recovery may fail dramatically. In this paper we propose an algorithm inspired by the Median-of-Means (MOM). Our algorithm guarantees recovery for heavy-tailed data, even in the presence of outliers. Theoretically, our results show our novel MOM-based algorithm enjoys the same sample complexity guarantees as ERM under sub-Gaussian assumptions. Our experiments validate both aspects of our claims: other algorithms are indeed fragile and fail under heavy-tailed and/or corrupted data, while our approach exhibits the predicted robustness.

## 1  Introduction

Compressive or compressed sensing is the problem of reconstructing an unknown vector $x^* \in \mathbb{R}^n$ after observing $m < n$ linear measurements of its entries, possibly with added noise: $y = Ax^* + \eta$, where $A \in \mathbb{R}^{m \times n}$ is called the measurement matrix and $\eta \in \mathbb{R}^m$ is noise. Even without noise, this is an underdetermined system of linear equations, so recovery is impossible without a structural assumption on the unknown vector $x^*$. The vast literature [84, 37, 72, 9, 18, 27, 2, 86, 11] on this subject typically assumes that the unknown vector is "natural," or "simple," in some application-dependent way.

Compressed sensing has been studied on a wide variety of structures such as sparse vectors [19], trees [20], graphs [90], manifolds [21, 89] or deep generative models [15]. In this paper, we concentrate on deep generative models, which were explored by [15] as priors for sample-efficient reconstruction. Theoretical results in [15] showed that if $x^*$ lies close to the range of a generative model $G : \mathbb{R}^k \to \mathbb{R}^n$ with $d-$layers, a variant of ERM can recover $x^*$ with $m = O(kd \log n)$ measurements. Empirically, [15] shows that generative models require $5 - 10\times$ fewer measurements to obtain the same reconstruction accuracy as Lasso. This impressive empirical performance has motivated significant recent research to better understand the behaviour and theoretical limits of compressed sensing using generative priors [36, 50, 62]

A key technical condition for recovery is the Set Restricted Eigenvalue Condition (S-REC) [15], which is a generalization of the Restricted Eigenvalue Condition [14, 17] in sparse recovery. This

condition is satisfied if $A$ is a sub-Gaussian matrix and the measurements satisfy $y = Ax^* + \eta$. This leads to the question: can the conditions on $A$ be weakened, and can we allow for outliers in $y$ and $A$? This has significance in applications such as MRI and astronomical imaging, where data is often very noisy and requires significant pruning/cleansing.

As we show in this paper, the analysis and algorithm proposed by [15] are quite fragile in the presence of heavy-tailed noise or corruptions in the measurements. In the statistics literature, it is well known that algorithms such as empirical risk minimization (ERM) and its variants are not robust to even a *single* outlier. Since the algorithm in [15] is a variant of ERM, it is susceptible to the same failures in the presence of heavy-tails and outliers. Indeed, as we show empirically in Section 6, precisely this occurs.

Importantly, recovery failure in the setting of [15] (which is also the focus of this paper) can be pernicious, precisely because generative models (by design) output images in their range space, and for well-designed models, these have high perceptual quality. In contrast, when a classical algorithm like LASSO [84] fails, the typical failure mode is the output of a non-sparse vector. Thus in the context of generative models, resilience to outliers and heavy-tails is especially critical. This motivates the need for algorithms that do not require strong assumptions on the measurements.

In this paper, we propose an algorithm for compressed sensing using generative models, which is robust to heavy-tailed distributions and arbitrary outliers. We study its theoretical recovery guarantees as well as empirical performance, and show that it succeeds in scenarios where other existing recovery procedures fail, without additional cost in sample complexity or computation.

## 1.1 Contributions

We propose a new reconstruction algorithm in place of ERM. Our algorithm uses a Median-of-Means (MOM) loss to provide robustness to heavy-tails and arbitrary corruptions. As S-REC may no longer hold, we necessarily use a different analytical approach. We prove recovery results and sample complexity guarantees for this setting even though previous assumptions such as the S-REC [15] condition do not hold. Specifically, our main contributions are as follows.

- (Algorithm) We consider robust compressed sensing for generative models where (i) a constant fraction of the measurements and measurement matrix are arbitrarily (perhaps maliciously) corrupted and (ii) the random ensemble only satisfies a weak moment assumption.

  We propose a novel algorithm to replace ERM. Our algorithm uses a median-of-means (MOM) tournament [65, 54] i.e., a min-max optimization framework for robust reconstruction. Each iteration of our MOM-based algorithm comes at essentially no additional computational cost compared to an iteration of standard ERM. Moreover, as our code shows, it is straightforward to implement.

- (Analysis and Guarantees) We analyze the recovery guarantee and outlier-robustness of our algorithm when the generative model is a $d$-layer neural network using ReLU activations. Specifically, in the presence of a constant fraction of outliers in $y$ and $A$, we achieve $\|G(\widehat{z}) - G(z^*)\|^2 \leq O(\sigma^2 + \tau)$ with sample size $m = O(kd \log n)$, where $\sigma^2$ is the variance of the heavy-tailed noise, and $\tau$ is the optimization accuracy. Using different analytical tools (necessarily, since we do not assume sub-Gaussianity), we show our algorithm, even under heavy-tails and corruptions, has the same sample complexity as the previous literature has achieved under much stronger sub-Gaussian assumptions. En route to our result, we also prove an interesting result for ERM: by avoiding the S-REC-based analysis, we show that the standard ERM algorithm does in fact succeed in the presence of a heavy-tailed measurement matrix, thereby strengthening the best-known recovery guarantees from [15]. This does not extend (as our empirical results demonstrate) to the setting of outliers, or of heavy-tailed measurement noise. For these settings, our new algorithm is required.

- (Empirical Support) We empirically validate the effectiveness of our robust recovery algorithm on MNIST and CelebA-HQ. Our results demonstrate that (as our theory predicts) our algorithm succeeds in the presence of heavy-tailed noise, heavy-tailed measurements, and also in the presence of arbitrary outliers. At the same time our experiments confirm that ERM can fail, and in fact fails dramatically: through an experiment on the CelebA-HQ data set, we demonstrate that the ERM recovery approach [15], as well as other natural approaches including $\ell_1$ loss minimization and trimmed loss minimization [81], can recover images that have little resemblance to the original.

## 1.2 Related work

Compressed sensing with outliers or heavy-tails has a long history. To deal with outliers only in $y$, classical techniques replace the ERM with a robust loss function such as $\ell_1$ loss or Huber loss [58, 74, 64, 24], and obtain the optimal statistical rates. Much less is known for outliers in $y$ and $A$ for robust compressed sensing. Recent progress on robust sparse regression [22, 10, 23, 26, 78, 60, 59, 81] can handle outliers in $y$ and $A$, but their techniques cannot be directly extended to arbitrary generative models $G$. Another line of research [43, 70, 65, 54] considers compressed sensing where the measurement matrix $A$ and $y$ have heavy-tailed distributions. Their techniques leverage variants of Median-of-Means (MOM) estimators on the loss function under weak moment assumptions instead of sub-Gaussianity, which generalize the classical MOM mean estimator in one dimension [73, 48, 3, 70].

[88] deals with compressed sensing of generative models when the measurements and the responses are non-Gaussian. However, the distribution model in [88] requires more stringent conditions compared to the weak moment assumption as will be specified in Definition 1, and their algorithm cannot tolerate arbitrary corruptions.

Generative priors have shown great promise in compressed sensing and other inverse problems, starting with [15], who generalized the theoretical framework of compressive sensing and restricted eigenvalue conditions [84, 27, 14, 17, 41, 13, 12, 28] for signals lying on the range of a deep generative model [33, 53]. Results in [50, 62, 47] established that the sample complexities in [15] are order optimal. The approach in [15] has been generalized to tackle different inverse problems [35, 8, 6, 71, 7, 79, 8, 61, 5, 46, 34, 4]. Alternate algorithms for reconstruction include [16, 25, 49, 30, 29, 82, 66, 25, 77, 38, 39]. The complexity of optimization algorithms using generative models have been analyzed in [32, 40, 57, 36]. See [75] for a more detailed survey on deep learning techniques for compressed sensing. A related line of work has explored learning-based approaches to tackle classical problems in algorithms and signal processing [1, 45, 69, 42].

## 2 Notation

For functions $f(n)$ and $g(n)$, we write $f(n) \lesssim g(n)$ to denote that there exists a universal constant $c_1 > 0$ such that $f(n) \leq c_1 g(n)$. Similarly, we write $f(n) \gtrsim g(n)$ to denote that there exists a universal constant $c_2 > 0$ such that $f(n) \geq c_2 g(n)$. We write $f(n) = O(g(n))$ to imply that there exists a positive constant $c_3$ and a natural number $n_0$ such that for all $n \geq n_0$, we have $|f(n)| \leq c_3 g(n)$. Similarly, we write $f(n) = \Omega(g(n))$ to imply that there exists a positive constant $c_4$ and a natural number $n_1$ such that for all $n \geq n_1$, we have $|f(n)| \geq c_4 g(n)$.

## 3 Problem formulation

Let $x^* = G(z^*) \in \mathbb{R}^n$ be the fixed vector of interest. The deep generative model $G : \mathbb{R}^k \to \mathbb{R}^n$ ($k \ll n$) maps from a low dimensional latent space to a higher dimensional space. In this paper, $G$ is a feedforward neural network with ReLU activations and $d$ layers.

Our definition of heavy-tailed samples assumes that the measurement matrix $A$ only has bounded fourth moment. Our corruption model is Huber's $\epsilon$-contamination model [44]. This model allows corruption in the measurement matrix $A$ and measurements $y$. Precisely, these are:

**Definition 1** (Heavy-tailed samples). *We say that a random vector $a$ is heavy-tailed if for a universal constant $C > 0$, the $4^{th}$ moment of $a$ satisfies*

$$\left(\mathbb{E}\left[\langle a, u \rangle^4\right]\right)^{\frac{1}{4}} \leq C \left(\mathbb{E}\left[\langle a, u \rangle^2\right]\right)^{\frac{1}{2}}, \qquad \forall u \in \mathbb{R}^n.$$

*For all $\delta > 0$, the $(4+\delta)^{th}$ moment of $a$ need not exist, and we make no assumptions on them.*

**Definition 2** ($\epsilon$-corrupted samples). *We say that a collection of samples $\{y_i, a_i\}$ is $\epsilon$-corrupted if they are i.i.d. observations drawn from the mixture*

$$\{y_i, a_i\} \sim (1-\epsilon)P + \epsilon Q,$$

*where $P$ is the uncorrupted distribution, $Q$ is an arbitrary distribution.*

Thus we assume that samples $\{y_i, a_i\}_{i=1}^m$ are generated from $(1 - \epsilon)P + \epsilon Q$, where $Q$ is an adversary, and $P$ satisfies the following:

**Assumption 1.** *Samples $(y_i, a_i) \sim P$ satisfy $y_i = a_i^\top G(z^*) + \eta_i$, where the random vector $a_i$ is isotropic and heavy-tailed as in Definition 2, and the noise term $\eta_i$ is independent of $a_i$, i.i.d. with zero mean and bounded variance $\sigma^2$.*

## 4    Our Algorithm

$\|\cdot\|$ refers to $\ell_2$ unless specified otherwise. The procedure proposed by [15] finds a reconstruction $\hat{x} = G(\hat{z})$, where $\hat{z}$ solves:

$$\widehat{z} := \underset{z \in \mathbb{R}^k}{\arg\min} \|AG(z) - y\|^2.$$

This is essentially an ERM-based approach. As is well known from the classical statistics literature, ERM's success relies on strong concentration properties, guaranteed, e.g., if the data are all sub-Gaussian. ERM may fail, however, in the presence of corruption or heavy-tails. Indeed, our experiments demonstrate that in the presence of outliers in $y$ or $A$, or heavy-tailed noise in $y$, [15] fails to recover $G(z^*)$.

**Remark** *Unlike typical problems in $M$-estimation and high dimensional statistics, the optimization problem that defines the recovery procedure here is non-convex, and thus in the worst case, computationally intractable. Interestingly, despite non-convexity, as demonstrated in [15], (some appropriate version of) gradient descent is empirically very successful. In this paper, we take this as a computational primitive, thus sidestepping the challenge of proving whether a gradient-descent based method can efficiently provide guaranteed inversion of a generative model. Our theoretical guarantees are therefore statistical but our experiments show empirically excellent performance.*

### 4.1    MOM objective

It is well known that the median of means estimator achieves nearly sub-Gaussian concentration for one dimensional mean estimation of variance bounded random variables [73, 48, 3]. Inspired by the median-of-means algorithm, we propose the following algorithm to handle heavy-tails and outliers in $y$ and $A$. We partition the set $[m]$ into $M$ disjoint batches $\{B_1, B_2, \cdots, B_M\}$ such that each batch has cardinality $b = \frac{m}{M}$. Without loss of generality, we assume that $M$ exactly divides $m$, so that $b$ is an integer. For the $j^{th}$ batch $B_j$, define the function

$$\ell_j(z) := \frac{1}{b}\|A_{B_j}G(z) - y_{B_j}\|^2, \tag{1}$$

where $A_{B_j} \in \mathbb{R}^{b \times n}$ denotes the submatrix of $A$ corresponding to the rows in batch $B_j$. Similarly, $y_{B_j} \in \mathbb{R}^b$ denotes the entries of $y$ corresponding to the batch $B_j$. Our workhorse is a novel variant of median-of-means (MOM) tournament procedure [65, 54] using the loss function eq. (1):

$$\widehat{z} = \arg\min_{z \in \mathbb{R}^k} \max_{z' \in \mathbb{R}^k} \underset{1 \leq j \leq M}{\text{median}}(\ell_j(z) - \ell_j(z')). \tag{2}$$

We do not assume that the minimizer is unique, since we only require a reconstruction $G(\widehat{z})$ which is close to $G(z^*)$. Any value of $z$ in the set of minimizers will suffice. The intuition behind this aggregation of batches is that if the inner player $z'$ chooses a point close to $z^*$, then the outer player $z$ must also choose a point close to $z^*$ in order to minimize the objective. Once this happens, there is no better option for $z'$. Hence a neighborhood around $z^*$ is almost an equilibrium, and in fact there can be no neighborhood far from $z^*$ with such an equilibrium.

**Computational considerations.** The objective function eq. (2) is not convex and we use Algorithm 1 as a heuristic to solve eq. (2). In Section 6, we empirically observe that gradient-based methods are able to minimize this objective and have good convergence properties. Our main theorem guarantees that a small value of the objective implies a good reconstruction and hence we can certify reconstruction quality using the obtained final value of the objective.

## 5    Theoretical results

We begin with a brief review of the Restricted Eigenvalue Condition in standard compressed sensing and show that S-REC is satisfied by heavy-tailed distributions.

---

**Algorithm 1** Robust compressed sensing of generative models

---
1: **Input:** Data samples $\{y_j, a_j\}_{j=1}^m$.
2: **Output:** $G(\widehat{z})$.
3: **Parameters:** Number of batches $M$.

---
4: Initialize $z$ and $z'$.
5: **for** $t = 0$ to $T - 1$, **do**
6:     For each batch $j \in [M]$, calculate $\frac{1}{|B_j|}(\ell_j(z) - \ell_j(z'))$ by eq. (1).
7:     Pick the batch with the median loss $\underset{1 \leq j \leq M}{\text{median}}(\ell_j(z) - \ell_j(z'))$, and evaluate the gradient for $z$ and $z'$ using

    backpropagation on that batch.
    (i) perform gradient descent for $z$;
    (ii) perform gradient ascent for $z'$.
8: **end for**
9: Output the $G(\widehat{z}) = G(z)$.

---

## 5.1 Set-Restricted Eigenvalue Condition for heavy-tailed distributions

Most theoretical guarantees for compressed sensing rely on variants of the Restricted Eigenvalue Condition(REC) [14, 17] and the closest to our setting is the Set Restricted Eigenvalue Condition [15](S-REC). Formally, $A \in \mathbb{R}^{m \times n}$ satisfies S-REC$(S, \gamma, \delta)$ on a set $S \subseteq \mathbb{R}^n$ if for all $x_1, x_2 \in S$,

$$\|Ax_1 - Ax_2\| \geq \gamma\|x_1 - x_2\| - \delta.$$

While we can prove many powerful results using the REC condition, proving that a matrix satisfies REC typically involves sub-Gaussian entries in $A$. If we don't have sub-Gaussianity, proving REC requires a finer analysis. A recent technique called the *small-ball method* [67] requires significantly weaker assumptions on $A$, and can be used to show REC [67, 85] for $A$ satisfying Assumption 1. While this technique can be used for sparse vectors, we do not have a general understanding of what structures it can handle, since existing proofs make heavy use of sparsity.

We now show that a random matrix whose rows satisfy Assumption 1 will satisfy S-REC over the range of a generator $G : \mathbb{R}^k \to \mathbb{R}^n$ with high probability. This generalizes Lemma 4.2 in [15]– the original lemma required i.i.d. sub-Gaussian entries in the matrix $A$, whereas the following lemma only needs the rows to have bounded fourth moments.

**Lemma 5.1.** *Let $G : \mathbb{R}^k \to \mathbb{R}^n$ be a $d-$layered neural network with ReLU activations. Let $A \in \mathbb{R}^{m \times n}$ be a matrix with i.i.d. rows satisfying Definition 1. For any $\gamma < 1$, if $m = \Omega\left(\frac{1}{1-\gamma^2}kd\log n\right)$, then with probability $1 - e^{-\Omega(m)}$, for all $z_1, z_2 \in \mathbb{R}^k$, we have*

$$\frac{1}{m}\|AG(z_1) - AG(z_2)\|^2 \geq \gamma^2\|G(z_1) - G(z_2)\|^2.$$

This implies that the ERM approach of [15] still works when we only have a heavy-tailed measurement matrix $A$. However, as we show in our experiments, heavy-tailed noise in $y$ and outliers in $y, A$ will make ERM fail catastrophically. In order to solve this problem, we leverage the median-of-means tournament defined in eq. (2), and we will now show it is robust to heavy-tails and outliers in $y, A$.

## 5.2 Main results

We now present our main result. Theorem 5.5 provides recovery guarantees in terms of the error in reconstruction in the presence of heavy-tails and outliers, where $\widehat{z}$ is the (approximate) minimizer of eq. (2). First we show that the minimum value of the objective in eq. (2) is indeed small if there are no outliers.

**Lemma 5.2.** *Let $M$ denote the number of batches. Assume that the measurements $y$ and measurement matrix $A$ are drawn from the uncorrupted distribution satisfying Assumption 1. Then with probability $1 - e^{-\Omega(M)}$, the objective in Equation (2) satisfies*

$$\min_{z \in \mathbb{R}^k} \max_{z' \in \mathbb{R}^k} \underset{1 \leq j \leq M}{\text{median}}(\ell_{B_j}(z) - \ell_{B_j}(z')) \leq 4\sigma^2. \tag{3}$$

We now introduce Lemma 5.3 and Lemma 5.4, which control two stochastic processes that appear in eq. (2). We show that minimizing the objective in eq. (2) implies that you are close to the unknown

vector $G(z^*)$. Notice that since $z^*$ is one feasible solution of the inner maximization step of $z'$, we can consider $z' = z^*$. Now consider the difference of square losses in eq. (2), which is given by:

$$\ell_j(\widehat{z}) - \ell_j(z^*) = \frac{1}{b}\|A_{B_j}G(\widehat{z}) - y_{B_j}\|^2 - \frac{1}{b}\|A_{B_j}G(z^*) - y_{B_j}\|^2,$$

$$= \frac{1}{b}\|A_{B_j}(G(\widehat{z}) - G(z^*))\|^2 - \frac{2}{b}\eta_{B_j}^\top(A_{B_j}(G(\widehat{z}) - G(z^*))),$$

where the last line follows from an elementary arithmetic manipulation.

Assume we have the following bounds on a majority of batches:

$$\frac{1}{b}\|A_{B_j}(G(\widehat{z}) - G(z^*))\|^2 \gtrsim \|G(\widehat{z}) - G(z^*)\|^2, \tag{4}$$

$$-\frac{2}{b}\eta_{B_j}^\top(A_{B_j}(G(\widehat{z}) - G(z^*))) \gtrsim -\|G(\widehat{z}) - G(z^*)\|. \tag{5}$$

Since the objective is the median of the sum of the above terms, a small value of the objective implies that $\|G(\widehat{z}) - G(z^*)\|$ is small. We formally show these bounds in Lemma 5.3, Lemma 5.4.

**Lemma 5.3.** *Let $G : \mathbb{R}^k \to \mathbb{R}^n$ be a generative model from a $d$-layer neural network using ReLU activations. Let $A \in \mathbb{R}^{m \times n}$ be a matrix with i.i.d. uncorrupted rows satisfying Definition 1. Let the batch size $b = \Theta\left(C^4\right)$, let the number of batches satisfy $M = \Omega(kd\log n)$, and let $\gamma$ be a constant which depends on the moment constant $C$. Then with probability at least $1 - e^{-\Omega(m)}$, for all $z_1, z_2 \in \mathbb{R}^k$ there exists a set $J \subseteq [M]$ of cardinality at least $0.9M$ such that*

$$\frac{1}{b}\|A_{B_j}(G(z_1) - G(z_2))\|^2 \geq \gamma^2\|G(z_1) - G(z_2)\|^2, \forall j \in J.$$

**Lemma 5.4.** *Consider the setting of Lemma 5.3 with measurements satisfying $y = AG(z^*) + \eta$, where $y, A, \eta$ satisfy Assumption 1 with noise variance $\sigma^2$. For a constant batch size $b$ and number of batches $M = \Omega(kd\log n)$, with probability at least $1 - e^{-\Omega(m)}$, for all $z \in \mathbb{R}^k$ there exists a set $J \subseteq [M]$ of cardinality at least $0.9M$ such that*

$$\frac{1}{b}|\eta_{B_j}^T A_{B_j}(G(z) - G(z^*))| \leq \sigma\|G(z) - G(z^*)\|, \forall j \in J.$$

The above lemmas do not account for the $\epsilon$−corrupted samples in Definition 2. However, since the batch size is constant in both the lemmas, there exists a value of $\epsilon$ such that sufficiently many batches have no corruptions. Hence we can apply Lemma 5.3, Lemma 5.4 to these uncorrupted batches. Using these lemmas with a constant batch size $b$, we obtain Theorem 5.5. We defer its proof to Appendix E.

**Theorem 5.5.** *Let $G : \mathbb{R}^k \to \mathbb{R}^n$ be a generative model from a $d$-layer neural network using ReLU activations. There exists a (sufficiently small) constant fraction $\epsilon$ which depends on the moment constant $C$ in Definition 1 such that the following is true. We observe $m = O(kd\log n)$ $\epsilon$-corrupted samples from Definition 2, under Assumption 1. For any $z^* \in \mathbb{R}^k$, let $\widehat{z}$ minimize the objective function given by eq. (2) to within additive $\tau$ of the optimum. Then there exists a (sufficiently large) constant $c$, such that with probability at least $1 - e^{-\Omega(m)}$, the reconstruction $G(\hat{z})$ satisfies*

$$\|G(\widehat{z}) - G(z^*)\|^2 \leq c(\sigma^2 + \tau),$$

*where $\sigma^2$ is the variance of noise under Assumption 1.*

We briefly discuss the implications of Theorem 5.5, with regards to sample complexity and error in reconstruction.

**Sample Complexity.**  Our sample complexity matches that of [15] up to constant factors. This shows that the minimizer of eq. (2) in the presence of heavy-tails and outliers provides the same guarantees as in the case of ERM with sub-Gaussian measurements.

**Statistical accuracy and robustness.**  Let us analyze the error terms in our theorem. The term $\tau$ is a consequence of the minimization algorithm not being perfect, since it only reaches within $\tau$ of the true minimum. Hence it cannot be avoided. The term $\sigma^2$ is due to the noise in measurements. In the

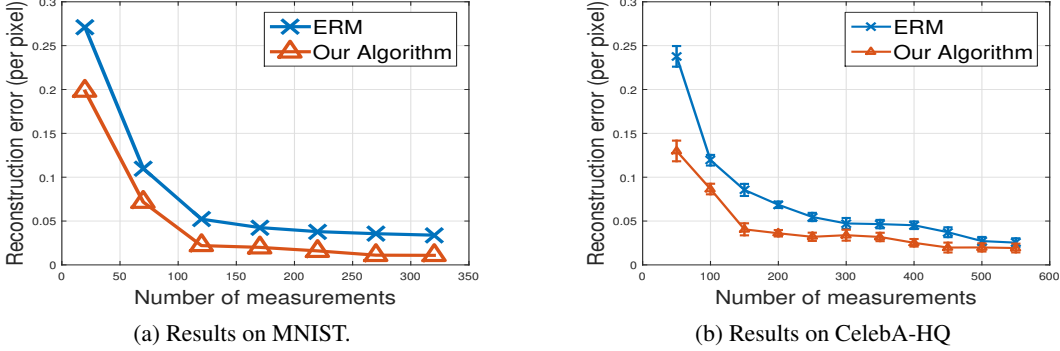

(a) Results on MNIST.  (b) Results on CelebA-HQ

Figure 1: We compare Algorithm 1 with the baseline ERM [15] under the heavy-tailed setting *without* arbitrary outliers. We fix $k = 100$ for the MNIST dataset and $k = 512$ for the CelebA-HQ dataset. We vary the number of measurements, and plot the reconstruction error per pixel averaged over multiple trials. With increasing number of measurements, we observe the reconstruction error decreases. For heavy-tailed $y$ and $A$ *without* arbitrary outliers, our method obtains significantly smaller reconstruction error in comparison to ERM.

main result of [15], the reconstruction $G(\widehat{z})$ has error bounded by $\|G(\widehat{z}) - G(z^*)\|^2 \lesssim \|\eta\|^2/m + \tau$.[2] This gives the following conditions:

- If $\eta$ is sub-Gaussian with variance $\sigma^2$, then $\|\eta\|^2/m \approx \sigma^2$ with high probability. Hence our bounds match up to constants.

- If higher order moments of $\eta$ do not exist, an application of Chebyshev's inequality says that with probability $1 - \delta$, [15] has $\|G(z^*) - G(\widehat{z})\|^2 \approx \sigma^2/(m\delta)$, and this can be extremely large if we want $\delta = e^{-\Omega(m)}$.

Hence our method is clearly superior if $\eta$ only has bounded variance, and if $\eta$ is sub-Gaussian, then our bounds match up to constants. In the presence of corruptions, [15] has no provable guarantee.

## 6 Experiments

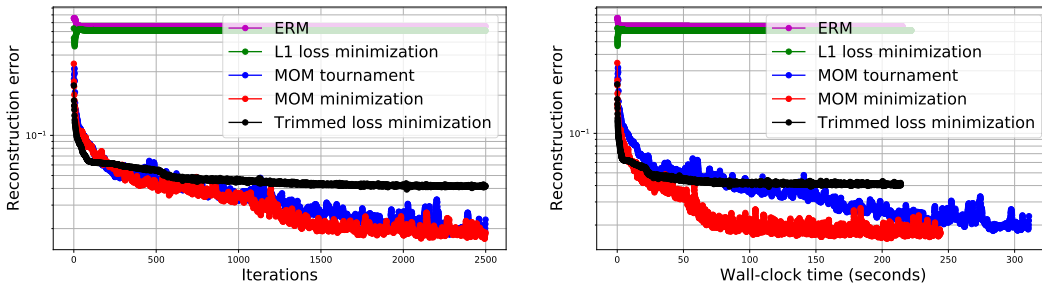

Figure 2: Plot of the reconstruction error versus the iteration number (left) and plot of the reconstruction error versus wall-clock time (right). ERM [15] and $\ell_1$ minimization fail to converge. Our two proposed methods, MOM tournament(blue) and MOM minimization(red), have the smallest reconstruction error. We provide a theoretical analysis for the MOM tournament algorithm, and observe that direct minimization of the MOM objective also works in practice. The computation time of our algorithms is nearly the same as the baselines.

In this section, we study the empirical performance of our algorithm on generative models trained on real image datasets. We show that we can reconstruct images under heavy-tailed samples and arbitrary outliers. For additional experiments and experimental setup details, see Appendix F.

**Heavy-Tailed Samples**  In this experiment, we deal with the *uncorrupted* compressed sensing model $P$, which has heavy-tailed measurement matrix and stochastic noise: $y = AG(z^*) + \eta$. We use

a Student's $t-$distribution (a typical example of heavy-tails) for $A$ and $\eta$. We compare Algorithm 1 with the baseline ERM [15] for heavy-tailed data *without* arbitrary corruptions on MNIST [55] and CelebA-HQ [51, 63]. We trained a DCGAN [80] with $k = 100$ and $d = 5$ layers to produce $64 \times 64$ MNIST images. For CelebA-HQ, we used a PG-GAN [51] with $k = 512$ to produce images of size $256 \times 256 \times 3 = 196,608$.

We vary the number of measurements $m$ and obtain the reconstruction error $\|G(\widehat{z}) - G(z^*)\|^2/n$ for Algorithm 1 and ERM, where $G(z^*)$ is the ground truth image. In Figure 1, Algorithm 1 and ERM both have decreasing reconstruction error per pixel with increasing number of measurements. To conclude, even for heavy-tailed noise *without* arbitrary outliers, Algorithm 1 obtains significantly smaller reconstruction error when compared to ERM.

**Arbitrary corruptions.** In this experiment, we use the same heavy-tailed samples as above, and we add $\epsilon = 0.02$-fraction of arbitrary corruption. We set the outliers of measurement matrix $A$ as random sign matrix, and the outliers of $y$ are fixed to be $-1$. We note that we don't use any targeted attack to simulate the outliers. We perform our experiments on the CelebA-HQ dataset using a PG-GAN of latent dimension $k = 512$, and fix the number of measurements to $m = 1000$.

We compare our algorithm to a number of natural baselines. Our first baseline is ERM [15] which is not designed to deal with outliers. While its fragility is interesting to note, in this sense it is not unexpected. For outliers in $y$, classical robust methods replace the loss function by an $\ell_1$ loss function or Huber loss function. This is done in order to avoid the squared loss, which makes recovery algorithms very sensitive to outliers. In this case, we have $\widehat{z} := \arg\min\|y - AG(z)\|_1$.

We also investigate the performance of trimmed loss minimization, which is a recent algorithm proposed by [81]. This algorithm picks the $t-$fraction of samples with smallest empirical loss for each update step, where $t$ is a hyper-parameter.

We run Algorithm 1 and its variant MOM minimization. The MOM minimization directly minimizes

$$\widehat{z} = \arg\min_{z \in \mathbb{R}^k} \text{median}_{1 \leq j \leq M}(\ell_j(z)), \tag{6}$$

and we use gradient-based methods similar to Algorithm 1 to solve it. Since Algorithm 1 optimizes $z$ and $z'$ in one iteration, the actual computation time of MOM tournament is twice that of MOM minimization. As shown in Figure 2, Figure 3, ERM [15] and $\ell_1$ loss minimization fail to converge to the ground truth and in particular, they may recover a completely different person. Trimmed loss minimization [81] only succeeds on occasion, and when it fails, it obtains a visibly different person. The convergence of the MOM minimization per iteration is very similar to the MOM tournament, and they both achieve much smaller reconstruction error compared to trimmed loss minimization. The right panel of Figure 2 plots the reconstruction error versus the actual computation time, showing our algorithms match baselines. We plot the MSE vs. number of measurements in Figure 4b, where the fraction of corruptions is set to $\epsilon = 0.02$.

**Miscellaneous Experiments** *Is ERM ever better than MOM?* So far we have analyzed cases where MOM performs better than ERM. Since ERM is known to be optimal in linear regression when dealing with uncorrupted sub-Gaussian data, we expect it to be superior to MOM when our measurements are all sub-Gaussian. We evaluate this in Fig. 4a and observe that ERM obtains smaller MSE in this setting. Notice that as we reduce the number of batches in MOM, it approaches ERM.

*How sensitive is MOM to the number of batches?* In Figure 4c we study the MSE of MOM tournaments and MOM minimization as we vary the number of batches.

In order to select the optimal number of batches ($M$), we keep a set of validation measurements that we do not use in the optimization routines for estimating $x$. We can run MOM for different value of $M$ to get multiple reconstructions, and then evaluate each reconstruction using the validation measurements to pick the best reconstruction. Note that one should use the median-of-means loss while evaluating the validation error as well.

## 7   Conclusion

The phenomenon observed in Figure 3 highlights the importance of our method. Our work raises several questions about why the objective we consider can be minimized, and suggests we need a new

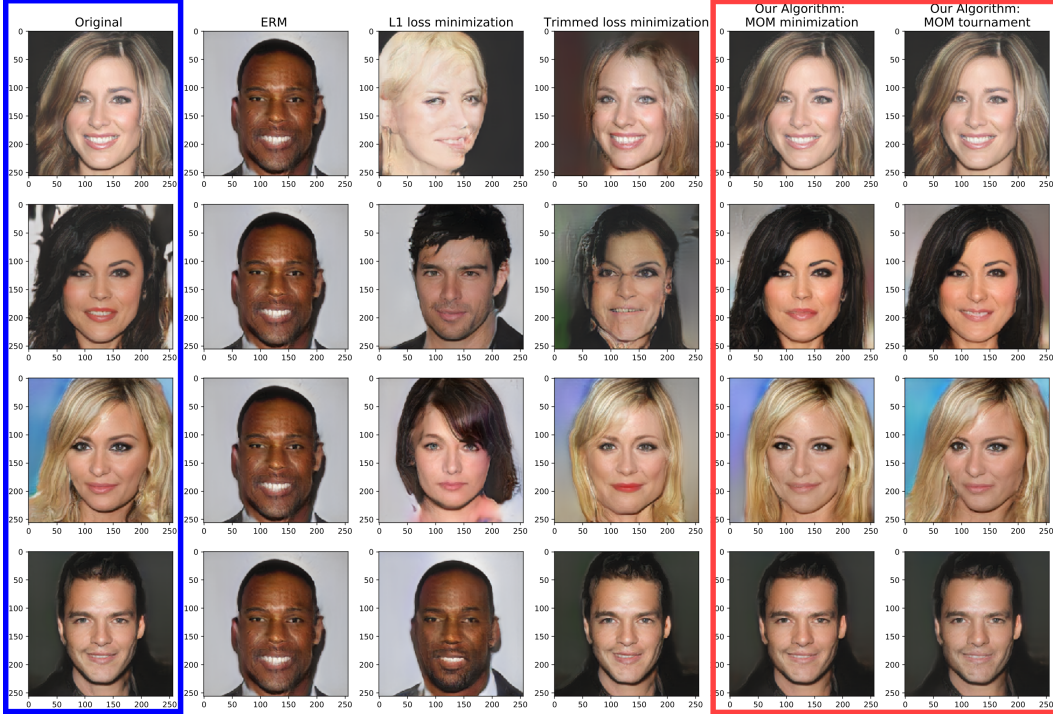

Figure 3: Reconstruction results on CelebA-HQ for $m = 1000$ measurements with 20 corrupted measurements. For each row, the first column is ground truth from a generative model. Subsequent columns show reconstructions by ERM [15], $\ell_1-$minimization, trimmed loss minimization [81]. In particular, vanilla ERM, $\ell_1-$minimization obtain completely different faces. Since we use the same outlier for different rows, vanilla ERM produces the same reconstruction irrespective of the ground truth. Trimmed loss minimization only succeeds on occasion (the last row), and when it fails, it obtains a similar but still different face. The last two columns show reconstructions by our proposed algorithms. The second to last one is directly minimizing the MOM objective eq. (6), and the last column minimizes the MOM tournament objective eq. (2). We provide a theoretical analysis for the MOM tournaments algorithm, and observe that direct minimization of the MOM objective also works in practice. We observe the last two columns have much better reconstruction performance – we get a high quality reconstruction under heavy-tailed measurements and arbitrary outliers.

paradigm for analysis that accounts for similar instances that enjoy empirical success, even though they can be provably hard in the worst case.

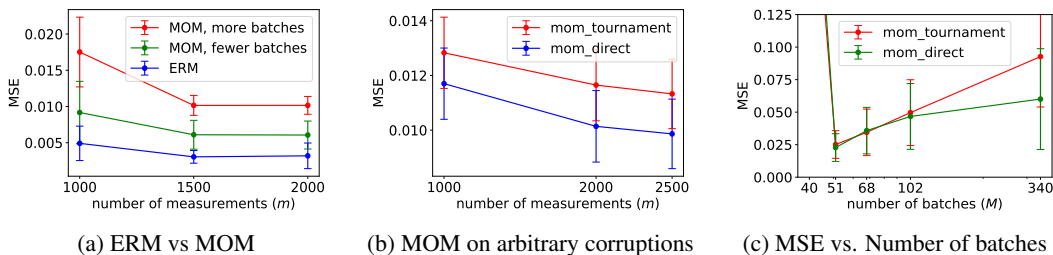

(a) ERM vs MOM      (b) MOM on arbitrary corruptions      (c) MSE vs. Number of batches

Figure 4: (a) We compare ERM and MOM by plotting MSE vs number of measurements when the measurements are *sub-Gaussian without corruptions*. (b) Aggregate statistics for MOM in the presence of corruptions. (c) MSE vs number of batches for MOM on 1000 heavy-tailed measurements and 20 corruptions. All error bars indicate 95% confidence intervals. Plots use a PGGAN on CelebA-HQ.

# 8 Acknowledgments

Ajil Jalal and Alex Dimakis have been supported by NSF Grants CCF 1763702,1934932, AF 1901292, 2008710, 2019844 research gifts by NVIDIA, Western Digital, WNCG IAP, computing resources from TACC and the Archie Straiton Fellowship. Constantine Caramanis and Liu Liu have been supported by NSF Award 1704778 and a grant from the Army Futures Command.

# 9 Broader Impact

Sparsity has played an important role across many areas of statistics, engineering and computer science, as a regularizing prior that captures important structure in many applications. Recent work has illustrated that given enough data, deep generative models are poised to play a revolutionary role, as a modern, data-driven replacement for sparsity. Much work remains to bring this agenda to fruition, but we believe that, as a variety of recent works have suggested, this direction can revolutionize imaging in a number of different important domains, not least of all, medical imaging.

This work addresses the robustness, and hence the trustworthiness and reliability of GAN-inversion-based techniques. As mentioned, this is especially critical, since high quality GANs will always produce perceptually high quality images, hence recovery failures may not be readily detectable by inspection.

Still, many significant issues remain that this work does not address. This includes understanding when and how sufficiently powerful and expressive GANs can be trained, since the scope of high quality GANs still appears to be limited. Another important consideration includes the core computational issue: the GAN inversion problem, which this work also faces, is intractable in the worst case, yet in practice appears to not pose a significant challenge. Understanding this dichotomy is very important.

## Footnotes

*Link to our code: https://github.com/ajiljalal/csgm-robust-neurips

[2]In [15], the bound is stated as $\|\eta\|^2$, but our $A$ has a different scaling, and hence the correct bound in our setting is $\|\eta\|^2/m$.

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
