[Supplementary Material]

# A Proof of Lemma 5.1

**Lemma** (Lemma 5.1). *Let $G : \mathbb{R}^k \to \mathbb{R}^n$ be a $d-$layered neural network with ReLU activations. Let $A \in \mathbb{R}^{m \times n}$ be a matrix with i.i.d rows satisfying Assumption 1. If $m = \Omega\left(\frac{1}{1-\gamma^2} kd \log n\right)$, then with probability $1 - e^{-\Omega(m)}$, $A$ satisfies*

$$\frac{1}{m}\|AG(z_1) - AG(z_2)\|^2 \geq \gamma^2 \|G(z_1) - G(z_2)\|^2$$

*for all $z_1, z_2 \in \mathbb{R}^k$.*

*Proof.* The proof is based on Proposition A.1 and Proposition A.2, which will be introduced as follows. Proposition A.1 shows that the set $S_G = \{G(z_1) - G(z_2) : z_1, z_2 \in \mathbb{R}^k\}$ lies in the range of $e^{O(kd \log n)}$ different $2k-$dimensional subspaces.

Proposition A.2 guarantees the result for a single subspace with probability $1 - e^{-m}$. Since $m = \Omega(kd \log n)$, the proof follows from a union bound over the $e^{O(kd \log n)}$ subspaces in Proposition A.1. $\square$

**Proposition A.1.** *If $G : \mathbb{R}^k \to \mathbb{R}^n$ is a $d-$layered neural network with ReLU activations, then the set $S_G = \{G(z_1) - G(z_2) : z_1, z_2 \in \mathbb{R}^k\}$ lies in the union of $O\left(n^{2kd}\right)$ different $2k-$dimensional subspaces.*

*Proof of Proposition* (A.1). From Lemma 8.3 in [15], the set $\{G(z) : z \in \mathbb{R}^k\}$ lies in the union of $O(n^{kd})$ different $k-$dimensional subspaces.

This implies that the set

$$\{G(z_1) - G(z_2) : z_1, z_2 \in \mathbb{R}^k\}$$

lies in the union of $M = O(n^{2kd})$ different $2k-$dimensional subspaces.

$\square$

**Proposition A.2.** *Consider a single $2k-$dimensional subspace given by $S_1 = \{Wz : W \in \mathbb{R}^{n \times 2k}, W^T W = I_{2k}, z \in \mathbb{R}^{2k}\}$. Let $A \in \mathbb{R}^{m \times n}$ be a matrix with i.i.d rows drawn from a distribution satisfying Assumption (1). If $m = O(\frac{C^2 k}{\frac{3}{4} - \gamma^2})$, with probability $1 - e^{-\Omega(m)}$, $A$ satisfies*

$$\frac{1}{m}\|Av\|^2 \geq \gamma^2 \|v\|^2, \ \forall v \in S_1.$$

*Proof.* The proof follows Theorem 14.12 in [87], with non-trivial modifications for our setting.

We want to show that for all vectors $v \in S_1$,

$$\frac{1}{m}||Av||^2 \geq \gamma^2 ||v||^2.$$

For $u, \tau \in \mathbb{R}$, define the truncated quadratic function

$$\phi_\tau(u) = \begin{cases} u^2 & \text{if } |u| \leq \tau, \\ \tau^2 & \text{otherwise.} \end{cases} \tag{7}$$

By construction, $\phi_\tau(\langle a_i, v \rangle) \leq \langle a_i, v \rangle^2$.

This implies that

$$\frac{1}{m}||Av||^2 = \frac{1}{m}\sum_{i=1}^{m}\langle a_i, v \rangle^2 = \frac{\|v\|^2}{m}\sum_{i=1}^{m}\langle a_i, \tfrac{v}{\|v\|} \rangle^2 \tag{8}$$

$$\geq \frac{\|v\|^2}{m}\sum_{i=1}^{m}\phi_\tau(\langle a_i, \tfrac{v}{\|v\|} \rangle) \tag{9}$$

$$\geq \|v\|^2 \mathbb{E}\left[\frac{\sum_{i=1}^{m}\phi_\tau(\langle a_i, \frac{v}{\|v\|}\rangle)}{m}\right] - \|v\|^2 \left|\frac{\sum_{i=1}^{m}\phi_\tau(\langle a_i, \frac{v}{\|v\|}\rangle)}{m} - \mathbb{E}\left[\frac{\sum_{i=1}^{m}\phi_\tau(\langle a_i, \frac{v}{\|v\|}\rangle)}{m}\right]\right| \tag{10}$$

$$= \|v\|^2 \mathbb{E}\left[\phi_\tau(\langle a, \tfrac{v}{\|v\|}\rangle)\right] - \|v\|^2 \left|\frac{\sum_{i=1}^{m}\phi_\tau(\langle a_i, \frac{v}{\|v\|}\rangle)}{m} - \mathbb{E}\left[\phi_\tau(\langle a, \tfrac{v}{\|v\|}\rangle)\right]\right| \tag{11}$$

$$\geq \|v\|^2 \mathbb{E}\left[\phi_\tau(\langle a, \tfrac{v}{\|v\|}\rangle)\right] - \|v\|^2 \sup_{v \in S_1}\left|\frac{1}{m}\sum_{i=1}^{m}\phi_\tau(\langle a_i, \tfrac{v}{\|v\|}\rangle) - \mathbb{E}\left[\phi_\tau(\langle a, \tfrac{v}{\|v\|}\rangle)\right]\right| \tag{12}$$

In Claim A.3 we will show that for $\tau^2 = \frac{C^4}{\frac{3}{4}-\gamma^2}$, we have

$$\mathbb{E}\left[\phi_\tau(\langle a, \tfrac{v}{\|v\|}\rangle)\right] \geq (\gamma^2 + \tfrac{1}{4}).$$

In Claim A.4 we will show that with overwhelming probability in $m$,

$$\sup_{v:\|v\|\leq 1}\left|\frac{1}{m}\sum_{i=1}^{m}\phi_\tau(\langle a_i, \tfrac{v}{\|v\|}\rangle) - \mathbb{E}\left[\phi_\tau(\langle a, \tfrac{v}{\|v\|}\rangle)\right]\right| \leq \frac{1}{4}.$$

These two results together imply that

$$\frac{1}{m}\|Av\|^2 \geq \gamma^2\|v\|^2.$$

with overwhelming probability in $m$. $\qquad\square$

**Claim A.3.** *Assume that the random vector $a$ satisfies Assumption (1) with constant $C$. Let $\phi_\tau$ be the thresholded quadratic function defined in Eqn (7). For all $v \in \mathbb{R}^n, \|v\| \leq 1$, we have*

$$\mathbb{E}\left[\phi_\tau(\langle a, v\rangle)\right] \geq \left(1 - \frac{C^4}{\tau^2}\right)\|v\|^2.$$

*Proof.*

$$\|v\|^2 - \mathbb{E}\left[\phi_\tau(\langle a, v\rangle)\right] = \mathbb{E}\left[\langle a, v\rangle^2\right] - \mathbb{E}\left[\phi_\tau(\langle a, v\rangle)\right] \tag{13}$$

$$= \mathbb{E}\left[(\langle a, v\rangle^2 - \tau^2)1_{\{|\langle a,v\rangle|\geq\tau\}}\right] \tag{14}$$

$$\leq \mathbb{E}\left[\langle a, v\rangle^2 1_{\{|\langle a,v\rangle|\geq\tau\}}\right] \tag{15}$$

By the Cauchy-Schwartz inequality,

$$\mathbb{E}\left[\langle a, v\rangle^2 1_{\{|\langle a,v\rangle|\geq\tau\}}\right] \leq \left(\mathbb{E}\left[\langle a, v\rangle^4\right]\right)^{\frac{1}{2}}\left(\Pr\left[|\langle a, v\rangle| \geq \tau\right]\right)^{\frac{1}{2}} \tag{16}$$

From Assumption (1), we have

$$\left(\mathbb{E}\left[\langle a, v\rangle^4\right]\right)^{\frac{1}{2}} \leq C^2\mathbb{E}\left[\langle a, v\rangle^2\right].$$

From Chebyshev's inequality and Assumption (1), we have

$$\left(\Pr\left[|\langle a, v\rangle| \geq \tau\right]\right)^{\frac{1}{2}} \leq \left(\frac{\mathbb{E}\left[|\langle a, v\rangle|^4\right]}{\tau^4}\right)^{\frac{1}{2}} \leq \left(\frac{C^4\mathbb{E}\left[|\langle a, v\rangle|^2\right]^2}{\tau^4}\right)^{\frac{1}{2}} = \frac{C^2\mathbb{E}\left[|\langle a, v\rangle|^2\right]}{\tau^2}. \tag{17}$$

Substituting the above two inequalities into eq. (16), we get

$$\mathbb{E}\left[\langle a, v\rangle^2 1_{\{|\langle a,v\rangle|\geq\tau\}}\right] \leq \frac{C^4\mathbb{E}\left[\langle a, v\rangle^2\right]^2}{\tau^2} \tag{18}$$

$$= \frac{C^4\|v\|^4}{\tau^2} \leq \frac{C^4\|v\|^2}{\tau^2}. \tag{19}$$

Substituting into Eqn (13),

$$\|v\|^2 - \mathbb{E}\left[\phi_\tau(\langle a, v \rangle)\right] \leq \frac{C^4 \|v\|^2}{\tau^2}, \tag{20}$$

which completes the proof. □

**Claim A.4.** *For an orthonormal matrix $U \in \mathbb{R}^{n \times 2k}$, let $S := \{v : v = Uz, \|v\| = 1\}$. Let $\phi_\tau$ be the function defined in Proposition A.2. For $m = \Omega\left(\tau^2 k\right)$, we have*

$$\sup_{v \in S} \left| \frac{1}{m} \sum_{i=1}^m \phi_\tau(\langle a_i, v \rangle) - \mathbb{E}\left[\phi_\tau(\langle a, v \rangle)\right] \right| \leq \frac{1}{4}.$$

*with probability $1 - e^{-\Omega(m)}$.*

*Proof.* Define

$$Z_m = \sup_{v \in S} \left| \frac{1}{m} \sum_{i=1}^m \phi_\tau(\langle a_i, v \rangle) - \mathbb{E}\left[\phi_\tau(\langle a, v \rangle)\right] \right|.$$

We will first show that

$$\mathbb{E}_A\left[Z_m\right] \leq \frac{1}{8}$$

for large enough $m$. Then we use Talagrand's inequality [83] to show that

$$\Pr\left[Z_m \geq \mathbb{E}\left[Z_m\right] + \frac{1}{8}\right] \leq e^{-\Omega(m)},$$

using which we can conclude that $Z_m \leq \frac{1}{4}$ with probability $1 - e^{-\Omega(m)}$.

By the symmetrization inequality, we have

$$\mathbb{E}_A\left[Z_m\right] \leq 2\mathbb{E}_{\epsilon, A}\left[\sup_{v \in S} \left| \frac{1}{m} \sum_{i=1}^m \epsilon_i \phi_\tau(\langle a_i, v \rangle) \right|\right]$$

where $\{\epsilon_i\}_{i=1}^m$ are i.i.d Bernoulli $\pm 1$ random variables.

Since $\phi_\tau$ is a Lipschitz function with Lipschitz constant $2\tau$, we can apply the Ledoux-Talagrand contraction inequality [56] (refer to Appendix G for the sake of completeness) to get

$$2\mathbb{E}_{\epsilon, A}\left[\sup_{v \in S} \left| \frac{1}{m} \sum_{i=1}^m \epsilon_i \phi_\tau(\langle a_i, v \rangle) \right|\right]$$

$$\leq 8\tau \mathbb{E}_{\epsilon, A}\left[\sup_{v \in S} \left| \frac{1}{m} \sum_{i=1}^m \epsilon_i \langle a_i, v \rangle \right|\right] \tag{21}$$

$$= 8\tau \mathbb{E}_{\epsilon, A}\left[\sup_{v \in S} \left| \frac{1}{m} \epsilon^T A v \right|\right]. \tag{22}$$

Since $S := \{v : v = Uz, \|v\| = 1\}$, we have

$$8\tau \mathbb{E}_{\epsilon, A}\left[\sup_{v \in S} \left| \frac{1}{m} \epsilon^T A v \right|\right] \tag{23}$$

$$= 8\tau \mathbb{E}_{\epsilon, A}\left[\sup_{z: \|z\|=1} \left| \frac{8\tau}{m} \epsilon^T A U z \right|\right] \tag{24}$$

$$\leq \frac{8\tau}{m} \mathbb{E}_{\epsilon, A}\left[\|\epsilon^T A U\|_2\right] \tag{25}$$

$$\leq \frac{8\tau}{m} \sqrt{\mathbb{E}_{\epsilon, A}\left[\|\epsilon^T A U\|_2^2\right]} \tag{26}$$

The third line follows from the Cauchy-Schwartz inequality, and the fourth line follows from Jensen's inequality.

Notice that
$$\mathbb{E}_\epsilon \left[ \|\epsilon^T A U\|_2^2 \right] = \text{trace}(AUU^T A^T) = \text{trace}(U^T A^T A U)$$

Since $U^T U = I_{2k}$, we have
$$\mathbb{E}_{\epsilon,A} \left[ \|\epsilon^T A U\|_2^2 \right] = \mathbb{E}_A \left[ \text{trace}(U^T A^T A U) \right] \tag{27}$$
$$= \sum_{i=1}^m \mathbb{E}_{a_i} \text{trace}(U^T a_i a_i^T U) \tag{28}$$
$$= \sum_{i=1}^m \text{trace}(U^T I_n U) = m \, \text{trace}(I_{2k}) = 2km. \tag{29}$$

Putting this together, and choosing $m = \Omega(\tau^2 k)$, we have
$$\mathbb{E}_A \left[ Z_m \right] \le 8\tau \sqrt{\frac{2k}{m}} \le \frac{1}{8}.$$

We now need to show that
$$\Pr \left[ Z_m \ge \mathbb{E} \left[ Z_m \right] + \frac{1}{8} \right] \le e^{-\Omega(m)}.$$

By construction, $\phi_\tau(\langle a_i, v \rangle) \le \tau^2$ for all $v \in S$.

In order to apply Talagrand's inequality, we need to bound
$$\sigma^2 = \sup_{v \in S} \mathbb{E} \left[ \left( \phi_\tau(\langle a, v \rangle) - \mathbb{E} \left[ \phi_\tau(\langle a, v \rangle) \right] \right)^2 \right].$$

We can bound this by
$$\text{var}(\phi_\tau(\langle a, v \rangle)) \le \mathbb{E} \left[ \phi_\tau^2(\langle a, v \rangle) \right] \tag{30}$$
$$\le \tau^2 \mathbb{E} \left[ \phi_\tau(\langle a, v \rangle) \right] \le \tau^2 \tag{31}$$

Applying Talagrand's inequality, we have
$$\Pr \left[ Z_m \ge \mathbb{E} \left[ Z_m \right] + t \right] \le C_1 \exp \left( -\frac{C_2 m t^2}{\tau^2 + \tau^2 t} \right).$$

Setting $t = \frac{1}{8}, m = \Omega(\tau^2 k)$ we get
$$\Pr[Z_m \ge \frac{1}{4}] \le \Pr \left[ Z_m \ge \mathbb{E} \left[ Z_m \right] + \frac{1}{8} \right] \le C_1 e^{-\frac{C_2 m}{\tau^2}} = e^{-\Omega(m)}.$$

This concludes the proof. $\qquad \square$

## B    Proof of Lemma 5.2

**Lemma B.1.** *Let $M$ denote the number of batches. Then with probability $1 - e^{-\Omega(M)}$, the objective in Equation (2) satisfies*
$$\min_{z \in \mathbb{R}^k} \max_{z' \in R^k} \text{median}_{1 \le j \le M} \ell_{B_j}(z) - \ell_{B_j}(z') \le 4\sigma^2. \tag{32}$$

*Proof.* By setting $z \leftarrow z^*$, for all $z' \in \mathbb{R}^k$, for any $j \in [M]$, we have
$$\ell_{B_j}(z^*) - \ell_{B_j}(z') \le \ell_{B_j}(z^*) = \frac{1}{b} \|\eta_{B_j}\|^2. \tag{33}$$

Since the noise is i.i.d. and has variance $\sigma^2$, we have $\mathbb{E}\left[\ell_{B_j}(z^*)\right] = \mathbb{E}\frac{1}{b}\|\eta_{B_j}\|^2 = \sigma^2$.

For batch $j \in [M]$, define the indicator random variable

$$Y_j = \mathbf{1}\left\{\ell_{B_j}(z^*) \geq 4\sigma^2\right\}.$$

By Markov's inequality, since $\mathbb{E}[\ell_{B_j}(z^*)] = \sigma^2$, we have

$$\Pr\left[Y_j = 1\right] \leq \frac{1}{4} \Rightarrow \mathbb{E}\left[\sum_{j=1}^{M} Y_j\right] \leq \frac{M}{4}. \tag{34}$$

By the Chernoff bound,

$$\Pr\left[\sum_{j=1}^{M} Y_j \geq \frac{M}{2}\right] \leq \Pr\left[\sum_{j=1}^{M} Y_j \geq 2\mathbb{E}[\sum_{j=1}^{M} Y_j]\right] \leq e^{-\Omega(M)}. \tag{35}$$

The above inequality implies that with probability $1 - e^{-\Omega(M)}$, for all $z' \in \mathbb{R}^k$, at least $\frac{M}{2}$ batches satisfy

$$\ell_{B_j}(z^*) - \ell_{B_j}(z') \leq 4\sigma^2.$$

This gives

$$\min_{z \in \mathbb{R}^k} \max_{z' \in R^k} \operatorname{median}_{1 \leq j \leq M}(\ell_{B_j}(z) - \ell_{B_j}(z')) \leq 4\sigma^2. \tag{36}$$

$\square$

## C  Proof of Lemma 5.3

**Lemma** (Lemma 5.3). *Let $G : \mathbb{R}^k \to \mathbb{R}^n$ be a generative model from a d-layer neural network using ReLU activations. Let $A \in \mathbb{R}^{m \times n}$ be a matrix with i.i.d rows satisfying Assumption 1. Let the batch size $b = \Theta\left(C^4\right)$, let the number of batches satisfy $M = \Omega(kd\log n)$, and let $\gamma$ be a constant which depends on the moment constant $C$. Then with probability at least $1 - e^{-\Omega(m)}$, for all $z_1, z_2 \in \mathbb{R}^k$ there exists a set $J \subseteq [M]$ of cardinality at least $0.9M$ such that*

$$\frac{1}{b}\|A_{B_j}(G(z_1) - G(z_2))\|^2 \geq \gamma^2\|G(z_1) - G(z_2)\|^2, \forall j \in J.$$

*Proof.* Proposition A.1 shows that the set $S_G = \{G(z_1) - G(z_2) : z_1, z_2 \in \mathbb{R}^k\}$ lies in the range of $e^{O(kd\log n)}$ different $2k-$dimensional subspaces.

Proposition C.1 guarantees the result for a single subspace with probability $1 - e^{-\Omega(M)}$. Since $M = \Omega(kd\log n)$ and the batch size is constant which depends on the moment constant $C$, the lemma follows from a union bound over the $e^{O(kd\log n)}$ subspaces in Proposition A.1.  $\square$

**Proposition C.1.** *Consider a single $2k-$dimensional subspace given by $S = \{Wz : W \in \mathbb{R}^{n \times 2k}, W^T W = I_{2k}, z \in \mathbb{R}^{2k}\}$. Let $A \in \mathbb{R}^{m \times n}$ be a matrix with i.i.d rows drawn from a distribution satisfying Assumption (1) with constant $C$. If the batch size $b = O(C^4)$ and the number of batches satisfies $M = \Omega\left(k\log\frac{1}{\epsilon}\right)$, with probability $1 - e^{-\Omega(M)}$, for all $x \in S$, there exist a subset of batches $J_x \subseteq [M]$ with $|J_x| \geq 0.90M$ such that*

$$\frac{1}{b}\|A_{B_j}x\|^2 \geq \gamma^2\|x\|^2 \,\forall j \in J_x,$$

*where $\gamma = \Theta\left(\frac{1}{C^2}\right)$ is a constant that depends on the moment constant $C$.*

*Proof.* Since the bound we want to prove is homogeneous, it suffices to show it for all vectors in $S$ that have unit norm. Let $W \in \mathbb{R}^{n \times 2k}$ be the orthonormal matrix spanning $S$, and $S_1$ denote the set of unit norm vectors in its span. That is,

$$S_1 = \{Wz : z \in \mathbb{R}^{2k}, \|z\| = 1, W \in \mathbb{R}^{n \times 2k}, W^T W = I_{2k}\}.$$

For a fixed $x \in S_1$ and $0 < t < 1$, we have

$$\mathbb{E}\left[\langle a, x \rangle^2\right] = \mathbb{E}\left[\langle a, x \rangle^2 \mathbf{1}\{\langle a, x \rangle \leq t^2 \|x\|^2\}\right] \mathbb{E}\left[\langle a, x \rangle^2 \mathbf{1}\{\langle a, x \rangle > t^2 \|x\|^2\}\right] \quad (37)$$

$$\leq t^2 \|x\|^2 + \mathbb{E}\left[\langle a, x \rangle^4\right]^{\frac{1}{2}} \left(\Pr\left[\langle a, x \rangle^2 \geq t^2 \|x\|^2\right]\right)^{\frac{1}{2}} \quad (38)$$

$$\leq t^2 \|x\|^2 + C^2 \|x\|^2 \left(\Pr\left[\langle a, x \rangle^2 \geq t^2 \|x\|^2\right]\right)^{\frac{1}{2}} \quad (39)$$

$$\Rightarrow \Pr\left[\langle a, x \rangle^2 \geq t^2 \|x\|^2\right] \geq \frac{\left(1 - t^2\right)^2 \|x\|^4}{C^4 \|x\|^4} = \frac{\left(1 - t^2\right)^2}{C^4} = C_1. \quad (40)$$

This is essentially a modified version of the Paley-Zigmund inequality [76].

Consider a batch $B_j$, which has $b$ samples. By the concentration of Bernoulli random variables, with probability $1 - 2e^{-\Omega(C_1 b)}$, we have

$$\sum_{i \in B_j} \mathbf{1}\left\{\langle a_i, x \rangle^2 \geq t^2 \|x\|^2\right\} \geq \frac{bC_1}{2}$$

This implies that if we set $b$ such that $1 - 2e^{-\Omega(C_1 b)} = 0.975$, then with probability 0.975, $B_j$ has $\frac{bC_1}{2}$ samples $\langle a_i, x \rangle$ whose magnitude is at least $t\|x\|$. This implies that the average square magnitude over the batch satisfies

$$\frac{1}{b}\|A_{B_j} x\|^2 = \frac{1}{b} \sum_{i \in B_j} \langle a_i, x \rangle^2 \geq t^2 \|x\|^2 \frac{bC_1}{2b} = \frac{C_1 t^2 \|x\|^2}{2}, \quad (41)$$

with probability 0.975.

Consider the indicator random variable associated with the complement of the above event. That is,

$$Y_j(x) = \left\{\frac{1}{b}\|A_{B_j} x\|^2 \leq \frac{C_1 t^2}{2} \|x\|^2.\right\}$$

From (41) we have that $\mathbb{E}\left[Y_j(x)\right] \leq 0.025$.

Consider the sum of indicator random variables over $M$ batches. By standard concentrations of Bernoulli random variables, we have with probabibility $1 - e^{-\Omega(M)}$,

$$\sum_{j=1}^{M} Y_j(x) \leq 2\mathbb{E}\left[\sum_{j=1}^{M} Y_j(x)\right] \leq 0.05.$$

This implies that there exist a subset of batches $J \subseteq [M]$ with $|J| \geq 0.95M$ such that

$$\frac{1}{b}\|A_{B_j} x\|^2 \geq \frac{C_1 t^2 \|x\|^2}{2} \quad \forall j \in J,$$

with probability $1 - e^{-\Omega(M)}$. This shows that we have the statement of the proposition for a fixed vector in $S_1$.

We now show that this holds true for an $\epsilon$-cover of $S_1$. Let $S_\epsilon$ denote a minimial $\epsilon$-covering of $S_1$. That is, $S_\epsilon$ is a finite subset of $S_1$ such that for all $x \in S_1$, there exists $\tilde{x} \in S_\epsilon$ such that $\|x - \tilde{x}\| \leq \epsilon$. Since $S_1$ has dimension $2k$ and diameter 1, we can find a set $S_\epsilon$ whose cardinality is at most $\left(O\left(\frac{1}{\epsilon}\right)\right)^{2k}$.

By a union bound, with probability $1 - e^{-\Omega(M)}|S_\epsilon|$, for all $\tilde{x} \in S_\epsilon$ there exists a subset of batches $J_{\tilde{x}} \subset [M]$ with $|J_{\tilde{x}}| \geq 0.95M$ such that

$$\frac{1}{b}\|A_{B_j} \tilde{x}\|^2 \geq \frac{C_1 t^2}{2} \quad \forall j \in J_{\tilde{x}} \quad (42)$$

Since $|S|_\epsilon \leq e^{O(k \log \frac{1}{\epsilon})}$, if $M = \Omega\left(k \log \frac{1}{\epsilon}\right)$, the above statement holds with probability $1 - e^{-\Omega(M)}$.

We now show that the statement of the proposition is true for all vectors in $S_1$. Since the proposition statement holds for an $\epsilon-$cover of $S_1$, we now only need to consider the effect of $A$ at a scale of $\epsilon$.

Now consider the set

$$S_2 = \{x - \tilde{x} : x \in S_1, \tilde{x} \in S_\epsilon, \|x - \tilde{x}\| \leq \epsilon\}.$$

Note that this a subset of all vectors in the span of $W$ that have norm at most $\epsilon$. That is, if

$$S_3 = \{Wz : z \in \mathbb{R}^{2k}, \|z\| \leq \epsilon\},$$

we have $S_2 \subseteq S_3$.

For a vector $v \in \mathbb{R}^n$, consider the random variable

$$Z_i(v) = \mathbf{1}\left[\langle a_i, v \rangle \geq \frac{\sqrt{C_1}t}{2\sqrt{2}}\right].$$

Define the random process

$$\Psi(a_1, a_2, \cdots, a_m) = \sup_{v \in S_2} \frac{1}{m} \sum_{i=1}^m \mathbf{1}\left[|\langle a_i, v \rangle| \geq \frac{\sqrt{C_1}t}{2\sqrt{2}}\right].$$

By the bounded difference inequality, with probability $1 - 2e^{-C_2\delta^2}$,

$$\Psi(a_1, a_2, \cdots, a_m) \leq \mathbb{E}\left[\Psi(a_1, a_2, \cdots, a_m)\right] + \frac{\delta}{\sqrt{m}}$$

Since $S_2 \subseteq S_3$, we can bound the expectation of $\Psi$ by

$$\mathbb{E}\left[\Psi(a_1, \cdots, a_m)\right] \leq \mathbb{E} \sup_{v \in S_3} \frac{1}{m} \sum_{i=1}^m \mathbf{1}\left[|\langle a_i, v \rangle| \geq \frac{\sqrt{C_1}t}{2\sqrt{2}}\right] \tag{43}$$

$$\leq \mathbb{E} \sup_{v \in S_3} \sum_{i=1}^m \frac{|\langle a_i, v \rangle|}{mt\sqrt{C_1}/2\sqrt{2}} \tag{44}$$

$$= \mathbb{E} \sup_{v \in S_3} \sum_{i=1}^m \frac{2\sqrt{2}|\langle a_i, v \rangle|}{mt\sqrt{C_1}} \tag{45}$$

$$\leq \mathbb{E} \sup_{v \in S_3} \left|\sum_{i=1}^m 2\sqrt{2}\frac{|\langle a_i, v \rangle| - \mathbb{E}\left[|\langle a, v \rangle|\right]}{mt\sqrt{C_1}}\right| + \sup_{v \in S_3} \sum_{i=1}^m \frac{2\sqrt{2}\mathbb{E}\left[|\langle a, v \rangle|\right]}{mt\sqrt{C_1}} \tag{46}$$

Since $a$ is isotropic and $v$ has norm at most $\epsilon$, by Jensen's inequality, we can bound the second term in the RHS by

$$\mathbb{E} \sup_{v \in S_3} \sum_{i=1}^m \frac{2\sqrt{2}\mathbb{E}\left[|\langle a, v \rangle|\right]}{mt\sqrt{C_1}} \lesssim \frac{\epsilon}{t\sqrt{C_1}}. \tag{47}$$

To bound the first term in the RHS, we use the Gine-Zinn symmetrization inequality [31, 68, 56]

$$\mathbb{E} \sup_{v \in S_3} \left|\sum_{i=1}^m 2\sqrt{2}\frac{|\langle a_i, v \rangle| - \mathbb{E}\left[|\langle a, v \rangle|\right]}{mt\sqrt{C_1}}\right| \lesssim \mathbb{E} \sup_{v \in S_3} \left|\sum_{i=1}^m \frac{\xi_i \langle a_i, v \rangle}{mt\sqrt{C_1}}\right| \tag{48}$$

where $\xi_i, i \in [m]$ are i.i.d $\pm 1$ Bernoulli variables.

We can bound this by

$$\mathbb{E} \sup_{v \in S_3} \left|\sum_{i=1}^m \frac{\xi_i \langle a_i, v \rangle}{mt\sqrt{C_1}}\right| = \mathbb{E}_{\xi,A}\left[\sup_{v \in S_3} \left|\frac{\xi^T A v}{mt\sqrt{C_1}}\right|\right], \tag{49}$$

$$= \mathbb{E}_{\xi,A}\left[\sup_{z:\|z\| \leq \epsilon} \left|\frac{\xi^T A W z}{mt\sqrt{C_1}}\right|\right] \tag{50}$$

$$\leq \mathbb{E}_{\xi,A}\left[\frac{\epsilon\|\xi^T AW\|}{mt\sqrt{C_1}}\right] \tag{51}$$

$$\leq \frac{\epsilon\sqrt{\mathbb{E}_{\xi,A}\|\xi^T AW\|^2}}{mt\sqrt{C_1}} \tag{52}$$

$$= \frac{\epsilon\sqrt{\mathbb{E}_A \text{trace}(AWW^T A^T)}}{mt\sqrt{C_1}} \tag{53}$$

$$= \frac{\epsilon\sqrt{2km}}{mt\sqrt{C_1}} \lesssim \frac{\epsilon}{t}\sqrt{\frac{k}{mC_1}} \tag{54}$$

The third line follows from the Cauchy-Schwartz inequality, and the fourth line follows from Jensen's inequality.

Since $m = Mb$, from the above inequality and Eqn (47) we can now bound $\mathbb{E}\Psi$ as

$$\mathbb{E}\left[\Psi(a_1,\cdots,a_m)\right] \lesssim \frac{\epsilon}{t}\sqrt{\frac{k}{MbC_1}} + \frac{\epsilon}{t\sqrt{C_1}} \tag{55}$$

Substituting the above inequality into the bounded difference inequality, we have with probability at least $1 - e^{-\Omega(\delta^2)}$,

$$\Psi(a_1,a_2,\cdots,a_m) \lesssim \frac{\epsilon}{t}\sqrt{\frac{k}{MbC_1}} + \frac{\epsilon}{t\sqrt{C_1}} + \frac{\delta}{\sqrt{Mb}} \tag{56}$$

Setting $M = \Omega(k), \delta = O\left(\sqrt{\frac{M}{b}}\right), \epsilon = O\left(\frac{t}{b}\sqrt{C_1}\right)$, we can reduce the terms in the above inequality to

$$\frac{\epsilon}{t}\sqrt{\frac{k}{MbC_1}} \leq O\left(\frac{1}{b^{\frac{3}{2}}}\right), \tag{57}$$

$$\frac{\epsilon}{t\sqrt{C_1}} \leq O\left(\frac{1}{b}\right), \tag{58}$$

$$\frac{\delta}{\sqrt{Mb}} \leq O\left(\frac{1}{b}\right), \tag{59}$$

Since $b > 1$, the sum of these three terms is dominated by $O\left(\frac{1}{b}\right)$. From this, we can conclude that for small enough $\epsilon, \delta$, with probability $1 - e^{-\Omega\left(\frac{M}{b}\right)}$,

$$\Psi(a_1,a_2,\cdots,a_m) \leq \frac{0.05}{b} \tag{60}$$

$$\Rightarrow \sup_{v\in S_3}\sum_{i=1}^{m}\mathbf{1}\left[|\langle a_i,v\rangle| \geq \frac{t\sqrt{C_1}}{2\sqrt{2}}\right] \leq 0.05M. \tag{61}$$

This allows us to control the effect of $A$ at a scale of $\epsilon$. It says that there at most $0.05M$ samples on which vectors with magnitude at most $\epsilon$ have a magnitude greater than $\frac{t\sqrt{C_1}}{2\sqrt{2}}$ after interacting with $A$. This implies that there at least $0.95M$ batches in which all samples are well behaved.

Since we have control over an $\epsilon-$cover of $S_1$ as well as vectors at a scale of $\epsilon$ in $S_1$, we can now prove our result for all vectors in $S_1$.

For any $x \in S_1$, let $\tilde{x} \in S_\epsilon$ be the point in the $\epsilon-$cover which is closest to $x$. For a batch $B_j$, we can express $\|A_{B_j}x\|$ as

$$\frac{1}{\sqrt{b}}\|A_{B_j}x\| \geq \frac{1}{\sqrt{b}}\|A_{B_j}\tilde{x}\| - \frac{1}{\sqrt{b}}\|A_{B_j}(x-\tilde{x})\|. \tag{62}$$

From (42), there exists a subset of batches $J_{\tilde{x}} \subseteq [M]$ with $|J_{\tilde{x}}| \geq 0.95M$ such that

$$\frac{1}{\sqrt{b}}\|A_{B_j}\tilde{x}\| \geq \frac{\sqrt{C_1}t}{\sqrt{2}} \ \forall\, j \in J_{\tilde{x}}. \tag{63}$$

From (61), there exists a subset of batches $J_{x-\tilde{x}} \subseteq [M]$ with $|J_{x-\tilde{x}}| \geq 0.95M$ such that for all $j \in J_{x-\tilde{x}}$,

$$|\langle a_i, x - \tilde{x}\rangle| \leq \frac{\sqrt{C_1}t}{2\sqrt{2}} \ \forall \ i \in B_j \tag{64}$$

$$\Rightarrow \frac{1}{\sqrt{b}}\|A_{B_j}(x - \tilde{x})\| \leq \frac{\sqrt{C_1}t}{2\sqrt{2}}, \tag{65}$$

$$\Rightarrow -\frac{1}{\sqrt{b}}\|A_{B_j}(x - \tilde{x})\| \geq -\frac{\sqrt{C_1}t}{2\sqrt{2}}. \tag{66}$$

From the bounds on $\|A_{B_j}\tilde{x}\|$ and the bound on $\|A_{B_j}(x - \tilde{x})\|$, we can conclude that for all $x \in S_1$ there exist a subset of batches $J_x = J_{\tilde{x}} \cap J_{x-\tilde{x}}$ with cardinality at least $0.9M$ such that

$$\frac{1}{\sqrt{b}}\|A_{B_j}x\| \geq \frac{\sqrt{C_1}t}{2\sqrt{2}}, \ \forall \ j \in J_x. \tag{67}$$

This completes the proof, with $\gamma = \frac{\sqrt{C_1}t}{2\sqrt{2}} = \frac{t(1-t^2)}{C^2 2\sqrt{2}}$. $\qquad\square$

# D  Proof of Lemma 5.4

**Lemma** (Lemma 5.4). *Consider the setting of Lemma 5.3 with measurements satisfying $y = AG(z^*) + \eta$. For any $t > 0$ and noise variance $\sigma^2$, let the batch size $b$ and number of batches $M$ satisfy $b = \Theta(\frac{\sigma^2}{t^2})$ and $M = \Omega(kd \log n)$. Then with probability at least $1 - e^{-\Omega(m)}$, for all $z \in \mathbb{R}^k$ there exists a set $J \subseteq [M]$ of cardinality at least $0.9M$ such that*

$$\frac{1}{b}|\eta_{B_j}^T A_{B_j}(G(z) - G(z^*))| \leq t\|G(z) - G(z^*)\| \ , \forall j \in J.$$

*Proof.* Proposition A.1 shows that the set $S_G = \{G(z_1) - G(z_2) : z_1, z_2 \in \mathbb{R}^k\}$ lies in the range of $e^{O(kd \log n)}$ different $2k-$dimensional subspaces. This trivially implies that for a fixed $z^* \in \mathbb{R}^k$, the set $\{G(z) - G(z^*) : z \in \mathbb{R}^k\}$ also lies in the range of $e^{O(kd \log n)}$ different $2k-$dimensional subspaces.

Proposition D.1 guarantees the result for a single subspace with probability $1 - e^{-\Omega(M)}$. Since $M = \Omega(kd \log n)$ and the batch size is constant which depends on the noise variance $\sigma^2$ and $t^2$, the lemma follows from a union bound over the $e^{O(kd \log n)}$ subspaces. $\qquad\square$

**Proposition D.1.** *Consider a single $2k-$dimensional subspace given by $S = \{Wz : W \in \mathbb{R}^{n \times 2k}, W^T W = I_{2k}, z \in \mathbb{R}^{2k}\}$. Let $A \in \mathbb{R}^{m \times n}$ be a matrix with i.i.d rows drawn from a distribution satisfying Assumption (1) with constant $C$. If the batch size $b = \Theta\left(\frac{\sigma^2}{t^2}\right)$ and the number of batches satisfies $M = \Omega\left(k \log \frac{1}{\epsilon}\right)$, with probability $1 - e^{-\Omega(M)}$, for all $x \in S$, there exist a subset of batches $J_x \subseteq [M]$ with $|J_x| \geq 0.90M$ such that*

$$\frac{1}{b}|\eta_{B_j}^T A_{B_j}x| \leq t\|x\| \ , \forall j \in J.$$

*Proof.* Since the bound we want to prove is homogeneous, it suffices to show it for all vectors in $S$ that have unit norm. Let $W \in \mathbb{R}^{n \times 2k}$ be the orthonormal matrix spanning $S$, and $S_1$ denote the set of unit norm vectors in its span. That is,

$$S_1 = \{Wz : z \in \mathbb{R}^{2k}, \|z\| = 1, W \in \mathbb{R}^{n \times 2k}, W^T W = I_{2k}\}.$$

Consider the set $S_\epsilon$, which is a minimal $\epsilon-$covering of $S_1$. That is, for every $x \in S_1$, there exists $\tilde{x} \in S_\epsilon$ such that $\|\tilde{x} - x\| \leq \epsilon$.

For a fixed $\tilde{x} \in S_\epsilon$, and $t > 0$, by Chebyshev's inequality,

$$\Pr\left[\frac{1}{b}|\eta^T A_{B_j}\tilde{x}| \geq \frac{t}{2}\right] \leq \frac{\sum_{i \in B_j}\left(\eta_i^2\langle a_i, \tilde{x}\rangle^2\right)}{b^2 t^2/4} \tag{68}$$

$$= \frac{b\sigma^2 \|\tilde{x}\|^2}{b^2 t^2 / 4} \tag{69}$$

$$= \frac{\sigma^2 4}{bt^2} \leq \frac{1}{40}, \tag{70}$$

if $b \geq \frac{160\sigma^2}{t^2}$.

Define the indicator random variable

$$Y_i(x) = \mathbf{1}\left\{\frac{1}{b}|\eta^T A_{B_i} x| \geq \frac{t}{2}\right\}.$$

From Eqn (70) we have

$$\mathbb{E}\left[Y_i(\tilde{x})\right] \leq \frac{1}{40}.$$

By concentration of Bernoulli variables, with probability $1 - e^{-\Omega(M)}$,

$$\sum_{j=1}^{M} Y_i(\tilde{x}) \leq 2\mathbb{E}\left[Y_1(\tilde{x})\right] \leq \frac{1}{20}.$$

This implies that for a fixed $\tilde{x} \in S_\epsilon$, with probability $1 - e^{-\Omega(M)}$, there exist a subset of batches $J_{\tilde{x}} \subseteq [M]$ with cardinality $0.95M$ such that

$$\frac{1}{b}|\eta^T A_{B_j} \tilde{x}| \leq \frac{t}{2} \ \forall \ j \in J_{\tilde{x}}. \tag{71}$$

Since the size of $S_\epsilon$ is at most $\left(O\left(\frac{1}{\epsilon}\right)\right)^{2k}$, we can union bound over all $\tilde{x}$ in $S_\epsilon$. Hence, if $M = \Omega\left(k \log \frac{1}{\epsilon}\right)$, then with probability $1 - e^{-\Omega(M)}$, for all $\tilde{x} \in S_\epsilon$, there exist a subset $J_{\tilde{x}} \subseteq [M]$ with cardinality $0.95M$ such that

$$\frac{1}{b}|\eta^T A_{B_j} \tilde{x}| \leq \frac{t}{2} \ \forall \ j \in J_{\tilde{x}}. \tag{72}$$

This shows that the multiplier component is well behaved on a large fraction of the batches for an $\epsilon-$cover of $S_1$. Now we need to extend the argument to all vectors in $S_1$.

Now consider the set
$$S_2 = \{x - \tilde{x} : x \in S_1, \tilde{x} \in S_\epsilon, \|x - \tilde{x}\| \leq \epsilon\}.$$

Note that this a subset of all vectors in the span of $W$ that have norm at most $\epsilon$. That is, if

$$S_3 = \{Wz : z \in \mathbb{R}^{2k}, \|z\| \leq \epsilon\},$$

we have $S_2 \subseteq S_3$.

For any $v \in \mathbb{R}^n$, define the random variable

$$Z_j(v) = \mathbf{1}\left\{|\eta_i a_i^T v| \geq \frac{t}{2}\right\}. \tag{73}$$

Now define the random process

$$\Psi(a_1, \cdots, a_m) = \sup_{v \in S_2} \frac{1}{m} \sum_{i=1}^{m} Z_i(v) \tag{74}$$

Since $S_2 \subseteq S_3$, we can bound $\mathbb{E}\left[\Psi\right]$ via

$$\mathbb{E}\left[\Psi\right] \leq \mathbb{E}\left[\sup_{v \in S_3} \frac{1}{m} \sum_{i=1}^{m} Z_i(v)\right] \tag{75}$$

$$\leq \mathbb{E}\left[\sup_{v \in S_3} \frac{1}{m} \sum_{i=1}^{m} \frac{|\eta_i a_i^T v|}{t/2}\right] \tag{76}$$

$$\leq \mathbb{E}\left[\sup_{v \in S_3} \left|\frac{1}{m} \sum_{i=1}^{m} \frac{|\eta_i a_i^T v| - \mathbb{E}|\eta_i a_i^T v|}{t/2}\right|\right]$$

$$+ \mathbb{E}\left[\sup_{v \in S_3} \frac{1}{m} \sum_{i=1}^{m} \frac{\mathbb{E}|\eta_i a_i^T v|}{t/2}\right] \tag{77}$$

We can bound the term on the right by

$$\mathbb{E}\left[\sup_{v \in S_3} \frac{1}{m} \sum_{i=1}^{m} \frac{\mathbb{E}|\eta_i a_i^T v|}{t/2}\right] \leq \frac{\mathbb{E}\left[\sup_{v \in S_3} \|\eta_i\|_2 \ |\langle a_i, v\rangle|\right]}{t/2} \tag{78}$$

$$\lesssim \frac{\sigma\epsilon}{t}, \tag{79}$$

where we have used the Cauchy Schwartz inequality, followed by the fact that $\eta$ is independent noise and has variance $\sigma^2$, $a$ is isotropic, and $v \in S_3$ has norm at most $\epsilon$.

To bound the term on the left, we use the Gine-Zinn symmetrization inequality [31, 68, 56]

$$\mathbb{E}\left[\sup_{v \in S_3} \left|\frac{1}{m} \sum_{i=1}^{m} \frac{|\eta_i a_i^T v| - \mathbb{E}|\eta_i a_i^T v|}{t/2}\right|\right] \lesssim \mathbb{E}\left[\sup_{v \in S_3} \left|\frac{1}{m} \sum_{i=1}^{m} \frac{\xi_i \eta_i a_i^T v}{t/2}\right|\right] \tag{80}$$

where $\xi_i, i \in [m]$ are i.i.d $\pm$ Bernoulli random variables.

Let $\xi\eta = (\xi_1\eta_1, \xi_2\eta_2, \cdots, \xi_m\eta_m)$ denote the the element wise product of the vectors $\xi = (\xi_1, \xi_2, \cdots, \xi_m)$ and $\eta = (\eta_1, \eta_2, \cdots, \eta_m)$. We can bound the above inequality by

$$\mathbb{E}\sup_{v \in S_3} \left|\sum_{i=1}^{m} \frac{\xi_i\eta_i\langle a_i, v\rangle}{mt/2}\right| = \mathbb{E}_{\xi,\eta,A}\left[\sup_{v \in S_3} \left|\frac{(\xi\eta)^T A v}{mt/2}\right|\right], \tag{81}$$

$$= \mathbb{E}_{\xi,\eta,A}\left[\sup_{z:\|z\|\leq\epsilon} \left|\frac{(\xi\eta)^T A W z}{mt/2}\right|\right] \tag{82}$$

$$\leq \mathbb{E}_{\xi,\eta,A}\left[\frac{\epsilon\|(\xi\eta)^T A W\|}{mt/2}\right] \tag{83}$$

$$\leq \frac{\epsilon\sqrt{\mathbb{E}_{\xi,\eta,A}\|(\xi\eta)^T A W\|^2}}{mt/2} \tag{84}$$

$$= \frac{\epsilon\sigma\sqrt{\mathbb{E}_A \text{trace}(A W W^T A^T)}}{mt/2} \tag{85}$$

$$= \frac{\epsilon\sigma\sqrt{2km}}{mt/2} \lesssim \frac{\epsilon\sigma}{t}\sqrt{\frac{k}{m}} \tag{86}$$

The third line follows from the Cauchy-Schwartz inequality, and the fourth line follows from Jensen's inequality, and the fifth line follows from the fact that $\xi\eta$ has i.i.d coordinates that are independent of $A$ and have variance $\sigma^2$.

From the above inequality and eq. (78), we get

$$\mathbb{E}[\Psi(a_1, a_2, \cdots, a_m)] \lesssim \frac{\sigma\epsilon}{t}\sqrt{\frac{k}{m}} + \frac{\sigma\epsilon}{t} \lesssim \frac{\sigma\epsilon}{t} \tag{87}$$

If we choose $\epsilon = c_1 \frac{t}{\sigma b}$ for a small enough constant $c_1$, then we can bound the expectation as

$$\mathbb{E}[\Psi(a_1, \cdots, a_m)] \leq \frac{0.025}{b} \tag{88}$$

By the bounded differences inequality, with probability $1 - e^{-\Omega(\delta^2)}$,

$$\Psi(a_1, \cdots, a_m) \leq \mathbb{E}\left[\Psi(a_1, \cdots, a_m)\right] + \frac{\delta}{\sqrt{m}} \tag{89}$$

Setting $\delta = 0.025\sqrt{\frac{M}{b}}$, we get $\frac{\delta}{\sqrt{m}} = \frac{0.025}{\sqrt{Mb}}\sqrt{\frac{M}{b}} = \frac{0.025}{b}$. This gives

$$\Psi(a_1, \cdots, a_m) \leq \frac{0.025}{b} + \frac{0.025}{b} = \frac{0.05}{b}. \tag{90}$$

From which we conclude that

$$\Rightarrow \sup_{v \in S_2} \sum_{i=1}^{m} \mathbf{1}\left\{|\eta_i a_i^T v| \geq \frac{t}{2}\right\} \leq \frac{0.05m}{b} = 0.05M. \tag{91}$$

Now consider any $x \in S_1$. There exists $\tilde{x} \in S_\epsilon$ such that $\|\tilde{x} - x\| \leq \epsilon$. From eq. (72) there exist a subset $J_{\tilde{x}} \subseteq [M]$ with cardinality $0.95M$ such that

$$\frac{1}{b}|\eta_{B_j}^T A_{B_j} \tilde{x}| \leq \frac{t}{2} \; \forall \, j \in J_{\tilde{x}}. \tag{92}$$

Similarly, from eq. (91), there exists a subset $J_{x-\tilde{x}} \subseteq [M]$ with cardinality $0.95M$ such that for all $j \in J_{x-\tilde{x}}$, we have

$$|\eta_i a_i^T(x - \tilde{x})| \leq \frac{t}{2} \; \forall \, i \in B_j, \tag{93}$$

$$\Rightarrow \frac{1}{b}|\eta_{B_j}^T A_{B_j}(x - \tilde{x})| \leq \frac{t}{2}. \tag{94}$$

From the triangle inequality and a simple union bound, for all $x \in S_1$, there exists a subset $J_x = J_{\tilde{x}} \cap J_{x-\tilde{x}}$ with cardinality $0.9M$ such that

$$\frac{1}{b}|\eta_{B_j}^T A_{B_j} x| \leq \frac{1}{b}|\eta_{B_j}^T A_{B_j}(x - \tilde{x})| + \frac{1}{b}|\eta_{B_j}^T A_{B_j} \tilde{x}| \tag{95}$$

$$\leq \frac{t}{2} + \frac{t}{2} = t \tag{96}$$

This completes the proof.

$\square$

# E    Proof of Theorem 5.5

*Proof.* In Theorem 5.5, we fix the batch size $b$ to be a suitable constant, specified in Lemma 5.3, Lemma 5.4. Then for $\epsilon \leq \frac{0.01}{b}$, the number of arbitrarily corrupted samples of $A$ and $y$ are at most $\frac{0.01}{b}bM = 0.01M$. This implies that there exist $0.99M$ batches with uncorrupted samples of $A, y$. For the rest of the proof, consider only these uncorrupted batches, and ignore the corrupted batches.

For a batch $j$, define the following

$$\mathbb{Q}_j(\hat{z}, z^*) := \frac{1}{b}\|A_{B_j}(G(\hat{z}) - G(z^*))\|^2, \tag{97}$$

$$\mathbb{M}_j(\hat{z}) := \frac{2}{b}\eta_{B_j}^\top(A_{B_j}(G(\hat{z}) - G(z^*))). \tag{98}$$

it is easy to verify that $\ell_j(\hat{z}) - \ell_j(z^*) = \mathbb{Q}_j(\hat{z}, z^*) - \mathbb{M}_j(\hat{z})$. The component $\mathbb{Q}_j(\hat{z}, z^*)$ is commonly called the quadratic component, and $\mathbb{M}_j(\hat{z})$ is called the multiplier component.

By Lemma 5.2, the minimum value of the MOM objective is at most $4\sigma^2$ with high probability. Since $\hat{z}$ minimizes the objective eq. (2) to within additive $\tau$ of the optimum, it implies that the median batch satisfies

$$\mathbb{Q}_j(\hat{z}, z^*) - \mathbb{M}_j(\hat{z}) \leq 4\sigma^2 + \tau. \tag{99}$$

Using Lemma 5.3, Lemma 5.4 on the $0.99M$ batches that do not have corruptions, if the batch size is a large enough constant, we see that there exist $0.78M$ batches on which both the following inequalities hold

$$\gamma^2 \|G(\widehat{z}) - G(z^*)\|^2 \leq \mathbb{Q}_j(\widehat{z}, z^*) \text{ and } -\sigma\|G(\widehat{z}) - G(z^*)\| \leq -\mathbb{M}_j(\widehat{z}). \qquad (100)$$

Putting the above two inequalities together, the median batch satisfies

$$\gamma^2 \|G(\widehat{z}) - G(z^*)\|^2 - \sigma\|G(\widehat{z}) - G(z^*)\| \leq 4\sigma^2 + \tau.$$

Solving the quadratic inequality for $\|G(\widehat{z}) - G(z^*)\|$, we have

$$\|G(\widehat{z}) - G(z^*)\|^2 \lesssim \sigma^2 + \tau. \qquad \square$$

## F  Experimental Setup

### F.1  MNIST dataset

We first compare Algorithm 1 with the baseline ERM [15] for heavy tailed dataset *without* arbitrary corruptions on MNIST dataset [55]. We trained a DCGAN [80] to produce $64 \times 64$ MNIST images.[3] We choose the dimension of the latent space as $k = 100$, and the model has 5 layers.

Based on this generative model, the uncorrupted compressed sensing model $P$ has heavy tailed measurement matrix and stochastic noise: $y = AG(z^*) + \eta$. We consider a Student's $t$ distribution (a typical example of heavy tails) – the measurement matrix $A$ is generated from a Student's $t$ distribution with degrees of freedom 4, and $\eta$ with degrees of freedom 3 with bounded variance $\sigma^2$. We vary the number of measurement $m$ and obtain the reconstruction error $\|G(\widehat{z}) - G(z^*)\|^2$ for Algorithm 1 and ERM, where $G(z^*)$ is the ground truth image. Each curve in Figure 1a demonstrates the averaged reconstruction error for 50 trials. In Figure 1a, Algorithm 1 and ERM both have decreasing reconstruction error per pixel with increasing number of measurement. In particular, Algorithm 1 obtains significantly smaller reconstruction error comparing with the baseline ERM.

### F.2  CelebA-HQ dataset

We continue the study of empirical performance of our algorithm on real image datasets with higher quality. We generate high quality RGB images with size $256 \times 256$ from CelebA-HQ[4]. Hence the dimension of each image is $256 \times 256 \times 3 = 196608$. In all of our experiments, we fix the dimension of the latent space as $k = 512$, and train a DCGAN on this dataset to obtain a generative model $G$.

We first compare our algorithm with the baseline ERM [15] for heavy tailed dataset without arbitrary corruptions, and then deal with the situation of outliers.

**Heavy tailed samples.**  In this experiment, we deal with the *uncorrupted* compressed sensing model $P$, which has heavy tailed measurement matrix and stochastic noise: $y = AG(z^*) + \eta$. We also use a Student's t distribution for $A$ and $\eta$ – the measurement matrix $A$ is generated from a Student's t distribution with degrees of freedom 4, and stochastic noise $\eta$ with degrees of freedom 3 with a bounded variance.

We obtain the reconstruction error $\|G(\widehat{z}) - G(z^*))\|$ vs. the number of measurement $m$ for our algorithm and ERM, where $z^*$ is the ground truth. In Figure 1b, each curve is an average of 20 trials. For heavy tailed $y$ and $A$ without any corruption, both methods are consistent, and have decaying reconstruction error with increasing sample size. Our method obtains significantly smaller reconstruction error, and shows competitive results over the baseline ERM for heavy tailed data set, even without any arbitrary outliers.

### F.3 Hyperparameter selection

When using the Adam [52] optimizer, we varied the learning rate over $[0.1, 0.05, 0.01, 0.005]$ for our algorithm and baselines. When using the Yellowfin [91] optimizer, we varied our learning rates over $[10^{-4}, 5 \cdot 10^{-5}, 10^{-5}, 5 \cdot 10^{-6}, 10^{-6}]$. We selected the best learning rate based on fresh measurements that were not used for optimization.

## G  Background

**Theorem G.1** (Ledoux-Talagrand Contraction Inequality)**.** *For a compact set $\mathcal{T}$, let $x_1, \cdots, x_m$ be i.i.d vectors whose real valued components are indexed by $\mathcal{T}$, i.e., $x_i = (x_{i,s})_{s \in \mathcal{T}}$. Let $\phi : \mathbb{R} \to \mathbb{R}$ be a 1-Lipschitz function such that $\phi(0) = 0$. Let $\epsilon_1, \cdots, \epsilon_m$ be independent Rademacher random variables. Then*

$$\mathbb{E}\left[ \sup_{s \in \mathcal{T}} \left| \sum_{i=1}^{m} \epsilon_i \phi(x_{i,s}) \right| \right] \leq 2\mathbb{E}\left[ \sup_{s \in \mathcal{T}} \left| \sum_{i=1}^{m} \epsilon_i x_{i,s} \right| \right].$$

**Theorem G.2** (Talagrand's Inequality for Bounded Empirical Processes)**.** *For a compact set $\mathcal{T}$, let $x_1, \cdots, x_m$ be i.i.d vectors whose real valued components are indexed by $\mathcal{T}$, i.e., $x_i = (x_{i,s})_{s \in \mathcal{T}}$. Assume that $\mathbb{E}x_{i,s} = 0$ and $|x_{i,s}| \leq b$ for all $s \in \mathcal{T}$. Let $Z = \sup_{s \in \mathcal{T}} \left| \frac{1}{m} \sum_{i=1}^{m} x_{i,s} \right|$. Let $\sigma^2 = \sup_{s \in \mathcal{T}} \mathbb{E}x_s^2$ and $\nu = 2b\mathbb{E}Z + \sigma^2$. Then*

$$\Pr\left[ Z \geq \mathbb{E}Z + t \right] \leq C_1 \exp\left( -\frac{C_2 m t^2}{\nu + bt} \right).$$

*where $C_1, C_2$ are absolute constants.*

## Footnotes

[3]Code was cloned from the following repository https://github.com/pytorch/examples/tree/master/dcgan.

[4]Code was cloned from the following repository: https://github.com/facebookresearch/pytorch_GAN_zoo.