[Reviews · NeurIPS 2020]

Review 1

Summary and Contributions: The paper proposes a median-of-means approach to increase the robustness to heavy-tailed and outlier measurement matrix errors in compressed sensing using generative models.

Strengths: The analysis and the theoretical results are sound. Experiments show the advantage of MoM over EMR.

Weaknesses: * Multiple approaches for compressive sensing with generative models exist in the literature and could be used in the numerical comparisons; the paper focuses on ERM. * The problem analyzed here does not seem to be commonly encountered - e.g, what settings provide this kind of error in the measurement matrix whereas a generative model is used for the data? The authors should provide a description of applications of interest, or better, yet, use data from these applications in their experiments. The experimental results appear to solve several toy problems.

Correctness: The claims and empirical methodology appear correct.

Clarity: * The paper is mostly clear. It seems that line 121 and after should be a definition separate from Definition 1? * The statement in line 154 “if the inner player z’ chooses a point close to z*” is imprecise. How close does the choice need to be, and how would this be achieved in practice? Similarly for the statement “a neighborhood around z* is almost an equilibrium” two lines later, and for the assumption “we have the following bounds on a majority of batches”. The authors have agreed to improve the presentation, which is welcome.

Relation to Prior Work: The analysis of robustness to heavy-tailed corruptions and outliers in the measurement matrices is new. The MoM method is commonly used and has been adapted to the CS setting.

Reproducibility: Yes

Additional Feedback: Small issues: Line 77: presence of heavy-tailed -> presence of a heavy-tailed Line 121: is a_i a row of A? Also, the distribution in the equation is for the entries of A? Line 126: Where is eta_i defined? Line 132: Gaussion -> Gaussian Line 191: In eq. (3), should ell_Bj be ell_j? Line 202: is this C the one from Definition 1?


Review 2

Summary and Contributions: This paper studies the problem of compressive sensing (recovery of images from under sampled linear measurements). Successful signal recovery requires a prior over the class of images being sought, and the paper considers priors given by generative models (such as GANs) that are mappings from a low dimensional latent space to image space. Existing approaches based on Empirical Risk Minimization have been shown to outperform classical techniques and to succeed under information-theoretically optimal sample complexity. This paper demonstrates that those methods fail in the presence of outliers, giving rise to the question of if there are efficient outlier-tolerant algorithms that can succeed at optimal sample complexity. This paper introduces a Median-of-Means (MOM) formulation for signal estimation from generative priors in the presence of outliers. The authors prove a recovery theorem that asserts that if the MOM algorithm is solved to within tau of the optimal value, if the measurements have heavy-tailed noise of standard deviation sigma, if sample complexity is proportional to the generator's latent dimensionality, if some constant fraction of measurements is arbitrarily corrupted, then the reconstructed images have error that scales like the sqrt(sigma^2 + tau). The authors provide a heuristic algorithm to approximate the solution to the MOM formulation and demonstrate lower reconstruction errors than ERM, an L1 loss, and another baseline. Contributions: - Introduction of MOM formulation for outlier robustness under generative priors. - Theorem with optimal sample complexity and recovery error for the MOM formulation. - Extension of existing results to heavy-tailed random measurement operators and noisy measurements. - Empirical demonstration of failure of existing approaches at outlier robustness and of strong performance of the MOM method.

Strengths: The strengths of this paper are: - Demonstrates that the default/naive method (L1 loss minimization under generative prior) for obtaining outlier robustness is insufficient for performing compressed sensing under generative priors. - Provides a novel (relative to this field) alternative approach for outlier robustness and provides compelling empirical demonstration of its better performance - Proves significant theoretical result for recovery performance using neural network priors, which is not that common in this field - The theoretical result is optimal in terms of sample complexity and scaling of recovery errors. - Demonstrates techniques for dealing with heavy-tailed measurement operators and measurement noise.

Weaknesses: The paper has no major weaknesses I can tell. Ideally, the authors would test their model on real data that contains outliers, but that is unnecessary given the primarily theoretical nature of this paper. There are critiques that could be applied to the philosophy of this approach (e.g. performance limitations of generative modeling, which has not yet been shown to be competitive in real-world applications). Nonetheless, this field deserves thorough exploration so that we can build methods down the road that eventually are competitive with non generative modeling approaches. Robustness is a major limitation of existing methods, and this paper helps address that.

Correctness: Yes

Clarity: Yes, but see "additional feedback" below for minor changes.

Relation to Prior Work: Yes

Reproducibility: Yes

Additional Feedback: Minor comments for improving the paper: (a) The authors claim in the abstract that their "algorithm guarantees recovery", but this is misleading because the theorem is not proved about the algorithm they write out, and the theory assumes that one can solve a potentially difficult optimization problem. The authors should be more upfront about this by conditioning their claim on the solvability of a minimax optimization. (The result is still beautiful, though readers should not get the impression that there is an algorithm with performance guarantees.) (b) They should explain the "L1 loss minimization" algorithm in their comparison plots. I presume this is the same as ERM but with L1-norm instead of L2-norm. The authors should correct me during rebuttal if I am wrong. (c) The authors should explain why the MOM tournament method is easier to analyze than the MOM minimization method. (d) The title of the paper is misleading. It is not doing Compressed Sensing *OF* Generative Models (which would mean recovering the generative model from compressive measurements); they are doing CS *WITH* Generative Models. (e) Line 9 of Algorithm 1: The notation z^T was not introduced or used anywhere else. (f) Lemma 4.1: What is gamma? Is a quantifier missing? (g) (4)-(5): Should define ">~ " notation. (h) Theorem 4.5: The authors should comment on what quantities depend on epsilon. (i) Theorem 4.5: The authors should remark that epsilon is small and could not, for example, be taken to be greater than 1/2. (j) (3): R^k -> \mathbb{R}^k (k) Definition 1: The choice of the term "universal constant C" is confusing here. Different measurement vectors would have different constants C, so C isn't "universal." ========== After Reading Response from Authors ============== I have read the authors response, and I continue to believe this is a good submission that should be accepted into this conference.


Review 3

Summary and Contributions: This paper provides an algorithm based on median-of-means (MOM) tournaments that recovers signals based on random measurements, where the signals can be compressed by some deep generative model. In contrast with classical methods such as ERM which only performs well with sub-gaussian measurements, the MOM algorithm can operate with heavy-tailed measurements. Moreover, since MOM algorithm and ERM share the same sample complexity, MOM algorithm can outperform ERM with same number of samples when the measurements are heavy-tailed.

Strengths: It is impressive that the authors are able to introduce a median-of-means tournaments algorithm that is computationally efficient. In contrast, original contributions were using median-of-means tournaments rather as a theoretical tool, but did not introduce computationally efficient version. The authors provide a good overview and plenty of references of the field. The presentation is clear since most of the technical details are postponed to appendix. The technical level of the proofs is very high. The experiment results are impressive and easy to read.

Weaknesses: There are some issues with notation and typos; for details see the suggestion section. There is little to no discussion of the drawbacks/limitation of the MOM algorithm, and the tone seems the suggest MOM is strictly better than ERM in all aspects, which will be very surprising to many readers. The authors might need to give more arguments/justifications for such statement, or clarify when/whether this is not the case.

Correctness: The methodologies for proof and experiments are reasonable. As far as I can see the proofs are correct.

Clarity: Yes. The presentation is very clear and the paper is very well organized.

Relation to Prior Work: The contribution is novel and clear.

Reproducibility: Yes

Additional Feedback: Some notation, although it might be conventional, should be defined explicitly: capital omega, theta, O for instance. Some notation is not acceptable, such as the mixture; the sum of two distributions is completely different from the composition of a random selector with two distribution. The pseudo code of the main algorithm should be made more compact, by representing sentences with equations and symbols. There are some typos that need to be fixed; the most obvious one is the equation above line 167. There is a missing summation in equation (35). One concern is with figure 3 where ERM performs horribly. Are there also cases where ERM outperforms MOM? If so it is also important to show those cases. -------------------- Comments after reading author feedback and other reviews: I still think that this is an excellent contribution. But I slightly adjusted my score in order to be more in line with the other reviewers.


Review 4

Summary and Contributions: The paper studies robust compressed sensing for generative models where the measurement matrix may be drawn from a heavy-tailed ensemble, and where the measurement matrix and measurements may be corrupted by arbitrary outliers. The authors give recovery guarantees for a tournament-style median-of-means algorithm, and an empirical evaluation demonstrates encouraging recovery results in practice over the range of GANs trained on the MNIST and CelebA-HQ datasets.

Strengths: 1. The problem formulation is clear and should be of interest to the NeurIPS community. Robust compressed sensing is an important and practically relevant area of research and the paper advances the understanding of CS over the range of generative models. 2. The proposed MOM algorithm is practical to implement and achieves encouraging results over standard datasets used for benchmarking GANs. 3. The theoretical analysis offers persuasive validation that approximately minimizing the objective is sufficient to guarantee reconstruction w.h.p. given a constant fraction of corrupted measurements. Moreover, Lemma 4.1 strengthens a previously-known result from [1] by allowing for heavy-tailed entries in the measurement matrix. [1] Bora, Jalal, Price, Dimakis. Compressed sensing using generative models. ICML 2017

Weaknesses: 1. Figure 1 should additionally indicate the standard deviation (or some similar measure of dispersion) of the datapoints. 2. For the evaluation of recovery under arbitrary corruptions, I would have liked to see aggregate statistics of the reconstruction error for each of the compared methods. For example, does MOM recovery consistently achieve good reconstruction, or is there some variance there? Does direct MOM minimization do just as well as the MOM tournament in practice? 3. It is perhaps somewhat surprising that the trimmed loss minimization heuristic performs as well as it does in Figure 2. Is it possible to achieve even better reconstruction by further fine tuning the hyperparameters of the method? Some text on how the baselines were tuned should be included. 4. How sensitive is the MOM algorithm to the selection of the number of batches M? There should be some indication of how M was chosen in Section 5 or the appendix.

Correctness: The claims and method appear to be correct, but I did not check the proofs in detail.

Clarity: The paper was very clearly written. The presentation of the theoretical results is helped along by the inclusion of useful intuition throughout Section 4.

Relation to Prior Work: I was satisfied with the coverage of prior work in the paper.

Reproducibility: Yes

Additional Feedback: Minor comments: 1. S4.1: S-REC definition should have x_1 - x_2 on RHS of the inequality. --- After author response: After reading the author response and the other reviews, I will continue to recommend acceptance. I would like to thank the authors for including the additional plots in the response, and for their detailed response to the question on the sensitivity to the number of batches.

[Author Response · NeurIPS 2020]



Figure 1: (a) MSE vs number of measurements when the measurements are *sub-Gaussian without corruptions*. (b) Plot adds confidence intervals for MSE in the heavy tailed setting *without corruptions*(Figure 1(b) in the original paper). (c) Aggregate statistics for MOM in the presence of corruptions. (d) MSE vs number of batches for MOM on 1000 heavy-tailed measurements and 20 corruptions. All error bars indicate 95% confidence intervals. Plots (a),(c),(d) use a PGGAN on CelebA-HQ while (b) uses a DCGAN on MNIST.

We thank the reviewers for their positive comments and useful suggestions. We are delighted that the reviewers found:
our problem well formulated & relevant; our theoretical results meaningful & novel; our experiments impressive &
practical. As noted (R2, R4), robust compressed sensing is an important and practically relevant problem. The proposed
Median-of-Means (MOM) algorithm advances the understanding of robust CS using generative models, in the setting
where measurements and measurement matrix are heavy-tailed and contain adversarial corruptions. We will modify the
notation and definition according to suggestions by R1, R2, R3, R4. We now address individual reviewer concerns.

@R3: Thank you for the positive feedback, we genuinely appreciate it. **"Are there cases where ERM outperforms**
**MOM?"** Yes – it is known that ERM is optimal when the measurements are subgaussian, with no adversarial
corruptions. We took this as a primitive and did not evaluate it in our submission. In Figure 1a, we plot MSE vs number
of measurements under *sub-Gaussian measurements without corruptions*. We believe the gap between ERM and MOM
is because the proposed objectives are harder to optimize, and can be reduced via fancier optimization routines that use
negative momentum. We will include this in future versions and conduct more extensive experiments to see if MOM
objectives are fundamentally harder to optimize. Note that as we reduce number of batches, MOM approaches ERM.

@R4: Thanks for the positive feedback and thought provoking questions. **Weakness 1: "Figure 1 should indicate**
**standard deviation"**. Please see Fig. 1b, which shows that our results are statistically significant. **Weakness 2: "Does**
**MOM consistently achieve good reconstruction? Does direct MOM minimization do just as well as the MOM**
**tournament in practice?"** Yes, they are consistent, and we find no statistically significant difference between them. In
Fig 1c we show MSE as the number of measurements are varied and the fraction of outliers is set to $0.02$. **Weakness**
**3: "Is it possible to achieve even better reconstruction by further fine tuning the hyperparameters of trimmed**
**loss minimization?"** We do not know – the difficulty with trimmed loss minimization is that we do not know how to
cross-validate hyperparameters. We were conservative and use $80\%$ of the samples at each gradient step. **Weakness**
**4: "How sensitive is MOM to the selection of the number of batches $M$?"** Excellent question, thank you for the
suggestion! In Fig 1d, we plot the MSE vs number of batches ($M$), under the setting of Section 5, Figure 3 in the main
paper (1000 heavy-tailed measurements + 20 corruptions). We find that the method fails when $M$ is too small, since the
majority of batches are corrupted. As long as $M$ is above a certain threshold, MSE increases slowly. In the main paper,
we used a conservative value of $M = 340$, as the qualitative results do not change much with $M$. Note that there exists
a cross-validation scheme for determining the optimal batch size, which we will include in the main paper due to space
constraints in this rebuttal.

@R2: Thanks for the positive feedback. We agree with your valuable suggestions and opinion that generative models
have limitations compared to classical approaches. However, given their constant progress, they have good potential to
be useful for real problems. On Question (b): you are correct, we will include a description for L1 minimization.

@R1: Thanks for the positive feedback. **"The problem does not seem to be commonly encountered [..] the**
**experimental results appear to solve several toy problems."** We respectfully disagree. We allow for outliers in the
*measurement matrix and measurement vector*. Robustness to measurement corruptions and outliers is fundamental for
compressed sensing theory (Comments from R2, R4 also support that). Practically, it can appear for mis-calibrated
medical imaging, malicious or strategic observations or other cases where robust statistics is relevant. @R1: **"The**
**paper focuses on ERM as a baseline despite other approaches for CS with generative models."** Indeed several
approaches exist for linear compressed sensing with generative models, but most of them suggest different algorithms
for approximately solving ERM (which is a non-convex and hard optimization problem). E.g., ADMM and PGD give
improvements over Bora et al. These algorithms will fail as they rely on ERM and do not consider robustness. The
only work on robust CS with generative models is by Wei et al. which does not propose a practical algorithm and
shows no experimental evaluation. Our work is the first practical algorithm for robust CS with generative models. @R1:
**"Statements (Ln. 154, 197, etc). are imprecise"** Indeed, we only want to build intution- we will try to improve the
presentation. Precise statements are in Lemma 4.3 & 4.4, and Theorem 4.5.

[Meta-Review · NeurIPS 2020]

The paper contains several contributions, ranging from the proposal of an algorithm to providing recovery guarantees, as well as experimental demonstration of the efficiency of the proposal. On the basis of the reviews and the author response, as well as my own reading of the paper, I would like to recommend acceptance of this paper.